# Mapping the functional connectivity of ecosystem services supply across a regional landscape

Rachel D Field*, Lael Parrott

The Okanagan Institute for Biodiversity, Resilience and Ecosystem Services (BRAES) Irving K. Barber Faculty of Science, Department of Biology, University of British Columbia, Kelowna, Canada

**Abstract** Sustainably managing multifunctional landscapes for production of multiple ecosystem services (ES) requires thorough understanding of the interactions between ES and the ecological processes that drive them. We build upon landscape connectivity theory to present a spatial approach for assessing functional connections between multiple ES at the landscape scale, and take a closer look at the concept of ES interactions by explicitly representing the mechanisms behind the relationships between ES. We demonstrate application of the approach using existing ES supply mapping data for plant agriculture, waterflow regulation, and landscape aesthetics and map the functional connectivity between them. We find that, when weights of all linkages were amalgamated, areas of high-value connectivity are revealed that are not present on any individual ES supply area or pairwise link maps, which suggests that the spatial focus of planning for optimal service provisioning may shift when functional relationships between several ES are considered. From water flow supply areas, our modeling maps several functional connections that operate over both short and long distances, which highlights the importance of managing ES flows both locally and across jurisdictions. We also found that different land use and land cover types than those associated with ES supply areas may be serving as critical corridors connecting interdependent ES. By providing spatial information on ES connectivity, our approach enables local and regional environmental planning and management to take full consideration of the complex, multi-scale interactions between ecological processes, land use and land cover, and ecosystem service supply on a landscape.

*For correspondence:
rdsfield@gmail.com

**Competing interest:** The authors declare that no competing interests exist.

## Editor's evaluation

Ecosystem services such as agriculture and waterflow regulation may interact, but the nature of these interactions is not well understood. This manuscript proposes a new framework based on approaches from geographic information science (GIS) to assess functional connectivity of ecosystem services, which reveals unexpected links across services and spaces. This paper is of interest to researchers in the fields of ecosystem services and landscape ecology, and more broadly to scientists studying sustainable practices affecting ecosystems.

## Introduction

Rapid human-driven modification of wilderness is placing increasing demands on landscapes to deliver nature's contributions to people, or 'ecosystem services' (ES; *Carpenter et al., 2009*). These juxtaposing forces highlight an urgent need for incorporating both biodiversity and ES in land use planning, with recent research calling specifically for consideration of landscape structure and connectivity in order to optimize environmental management objectives (*Mitchell et al., 2013*; *Ekroos et al.,*

*2014*; *Werling et al., 2014*; *Dobbs et al., 2014*). The boom in ES research over the past several decades has improved our understanding of the ecological drivers underpinning the supply of ES, but more nuanced work is necessary to meaningfully manage ES provision and their interdependencies at the landscape scale (*Kremen and Ostfeld, 2005*; *Tscharntke et al., 2005*; *Nicholson et al., 2009*; *Daily et al., 2009*). Specifically, the supply of an ES is typically mapped within fixed areas (e.g. *Tallis et al., 2008*) without considering the potential relevance of ecological process flux across the landscape for supporting ES provisioning (e.g. *Mitchell et al., 2013*) and multi-ES relationships. By failing to represent the spatial and functional connectivity between supply areas in ES assessment, we ignore ecological processes that may be fundamental to the maintenance of ES supplies, run the risk of overlooking potentially critical areas in landscape-scale management, and miss opportunities for uniting divergent interest groups around the concept of multifunctional landscapes (i.e. those that provide multiple ES beyond those that are primarily managed; *Power, 2010*). To optimize ES provisioning while minimizing potential negative effects on human well-being in the face of increased development pressures, it is critical to understand the dynamics of multi-ES supply (*Lorilla et al., 2018*).

Connectivity is a key attribute of landscape resilience and of ES in general (e.g. *Bennett et al., 2021*). Loss of connectivity through fragmentation or decreases in habitat, biotic and/or abiotic supplies can have deleterious effects on the wealth of biodiversity and natural capital and ultimately lead to declines in total ES supply and in the quantity and/or quality of flows to human beneficiaries (*Mitchell et al., 2015*; *Pal et al., 2021*). Landscape fragmentation impacts the supply of ES through altering the distribution and movement of the ecological elements, structures and processes underpinning the maintenance of natural capital (*Mitchell et al., 2015*). *Mitchell et al., 2015* discuss how loss of connectivity can be a driver of interactions between multiple ES and can impact both the size and location of ES flows (*Bagstad et al., 2013*). Among key policy principles identified for enhancing ES resilience to disturbances and environmental changes is managing for connectivity among ES-related resources, species, and human actors, with specific focus on the strength and structure of these connections (*Biggs et al., 2012*). All this points to the importance of planning for connectivity in multifunctional landscapes (*Phillips et al., 2015*), while considering the potential for complex ecological process-based interactions among services, to successfully manage for the delivery of multiple ES (*Dee et al., 2017*).

In simple terms, planning for landscape connectivity typically focuses on habitat patches and movement corridors, whereas ES planning focuses on the areas of the landscape with the capacity to produce the services humans need to survive and thrive (*Taylor et al., 1993*; *Egoh et al., 2008*). Recent work calling explicitly for incorporating ES into connectivity research has taken the perspective of assessing how the characteristics of landscape connectivity (i.e. how a landscape promotes or hinders movement of matter and organisms), along with composition (i.e. quantities of land use and land cover, or 'LULC', types), and configuration (i.e. spatial pattern of LULC), might directly or indirectly impact ES provision and related ecological processes (*Debinski and Holt, 2001*, *Fahrig, 2003*; *Gonzalez et al., 2009*; *Mitchell et al., 2013*). For example, adequate arrangement of adjacent natural habitat areas in agricultural landscapes can aid the movement of pollinators and pest predators to croplands, and thus promote delivery of these services (*Priess et al., 2007*; *Ricketts et al., 2008*; *Fleischner, 1994*; *Tscharntke and Brandl, 2004*; *Kremen et al., 2007*; *Tallis and Polasky, 2009*; *Power, 2010*; *Lonsdorf et al., 2011*). In terms of abiotic flows, connections between upstream and downstream freshwater sources can be important for maintaining quantity and quality of drinking water (*Dodds and Oakes, 2008*; *Bangash et al., 2013*), and maintenance of the natural hydrologic regime stabilizes base flows and reduces flooding, thereby promoting waterflow regulation (*Poff et al., 1997*). The above examples highlight that the ecological processes underpinning the supplies of certain ES directly influence the supplies of others, both when services co-occur in space and, sometimes, when they are produced in separate areas. Drivers behind multi-ES interactions, and the importance of such processes, are sometimes discussed in ES interaction research (e.g. *Li et al., 2017*; *Alemu et al., 2021*) but, to our knowledge, have not been explicitly delineated on the landscape in the context of multi-ES assessments.

Since the seminal global appraisal of ecosystems and the ES they provide (*Millennium Ecosystem Assessment, 2005*), research that assesses the *interactions* between multiple services has increased exponentially (*Agudelo et al., 2020*; Appendix 1). However, research in this discipline commonly only considers services that co-occur in space (e.g. *Queiroz et al., 2015*), and assumes positively

or negatively correlated ES to represent synergistic production or trade-offs, respectively (*Tomscha and Gergel, 2016*; *Agudelo et al., 2020*). Such assessments are typically based on correlation coefficients of indicators aggregated within a geographic unit (e.g. watershed, municipality) or randomly sampled across a region (*Anderson et al., 2009*; *Qin et al., 2015*; *Carpenter et al., 2015*). These approaches do not directly evaluate interactions based on underlying ecological process theory nor do they allow for spatially discrete relationships to occur, that is, they do not explicitly incorporate the mechanisms responsible for ES interactions, and they ignore how ES occurring in one area might have direct or indirect influence on ES in other areas. It has also been shown that simple spatial correlation analyses between pairs of ES are not necessarily a good predictor of how relationships between ES change over time (*Mitchell et al., 2020*), and that their interactions can vary across the LULC types found in heterogeneous landscapes (*Li et al., 2017*); thus, a better understanding of the processes that underpin the spatial patterns of ES is needed to improve the sustainable management of multi-functional landscapes (*Mitchell et al., 2020*). Recent research has visualized the spatial connectivity between ES supply areas by modeling the movement potential of species through high-quality habitat corridors as a proxy for how biodiversity flow in general supports ES provisioning across the landscape (*Peng et al., 2018*). Still, this does not represent different functional connections between ES supply, and how the provisioning of one type of ES directly or indirectly effects the provisioning of another across a landscape. Further, a recent systematic review of studies that model interactions among multiple ES between 2005 and 2019 found that the majority of studies were conducted locally while relatively few studies were done at the regional scale (*Agudelo et al., 2020*). However, focus at the regional level may be most appropriate for reconciling the common scale mismatches between biophysical and socio-economic elements involved in sustainable ES management (e.g. *Dalgaard et al., 2003*; *Cumming et al., 2006*; *Satake et al., 2008*; *Ingram et al., 2008*), while minimizing practical issues with empirical mapping related to data gaps and indicator variability in areas larger than this (*Verburg and Chen, 2000*; but see *de Groot et al., 2010* for examples of variability in ecological scale relevance for specific ES). As this relates to interactions between different ES, small-scale observations may be masked at larger scales (*Raudsepp-Hearne and Peterson, 2016*); therefore, incorporating local, grid-level data and analyses is also important for providing meaningful information to planners (*Haase et al., 2012*).

In spite of the growing knowledge around the complex interactions and feedbacks between ES, the related suite of biotic and abiotic mechanisms, and the cruciality of incorporating this into decision-making (*Qiu and Turner, 2013*; *Dee et al., 2017*), spatial modeling of the diverse functional connections between multiple ES (e.g. *Cui et al., 2012*; *Kolosz et al., 2018*; *Agudelo et al., 2020*) from several broad ES categories at the regional scale remains limited (*Field et al., 2017*). Several approaches used in ecological connectivity studies to identify potential spatial linkages across a landscape are promising in their applicability to multi-ES assessment. These include Euclidean distances (*Cressie, 1993*), least-cost path analysis (LCP; *Larkin et al., 2004*), least-cost corridor (LCC; *Singleton, 2002*), circuit theory (*McRae and Beier, 2007*), graph theory (*Pinto and Keitt, 2008*), and network flow models (*Phillips et al., 2008*). All these approaches are potentially amenable to assessment of multi-ES interactions but, to date, we know of no studies that have applied such methods to map the process-driven interactions between the supplies of multiple ES in a regional context (*Peng et al., 2018*). Further, studies that have incorporated both landscape connectivity and ES concepts typically only focus on a single ES, are skewed toward specific types of provisioning (e.g. food) and regulating (e.g. pollination) services, and, to our knowledge, have not yet tested cultural services (*Mitchell et al., 2013*).

The purpose of this study is to present an approach to address the above research gaps, building on existing ES mapping and modeling and rooted in landscape connectivity theory, where we identify relevant functional relationships between multiple ES and demonstrate how these can be spatially represented in the context of connectivity planning across a regional heterogeneous landscape. We demonstrate our approach using existing grid-level data from a case study landscape in the southern interior of British Columbia, Canada, by mapping and assessing the connectivity between ES from three broad categories: provisioning (plant food agriculture, PA), regulating (waterflow regulation, WF), and cultural (landscape aesthetics, LA; *Millennium Ecosystem Assessment, 2005*; *Figure 1*). Using these, we conceptualize ES supply areas as structural components, and the functional process links between these areas as configuration elements within a landscape connectivity framework. We

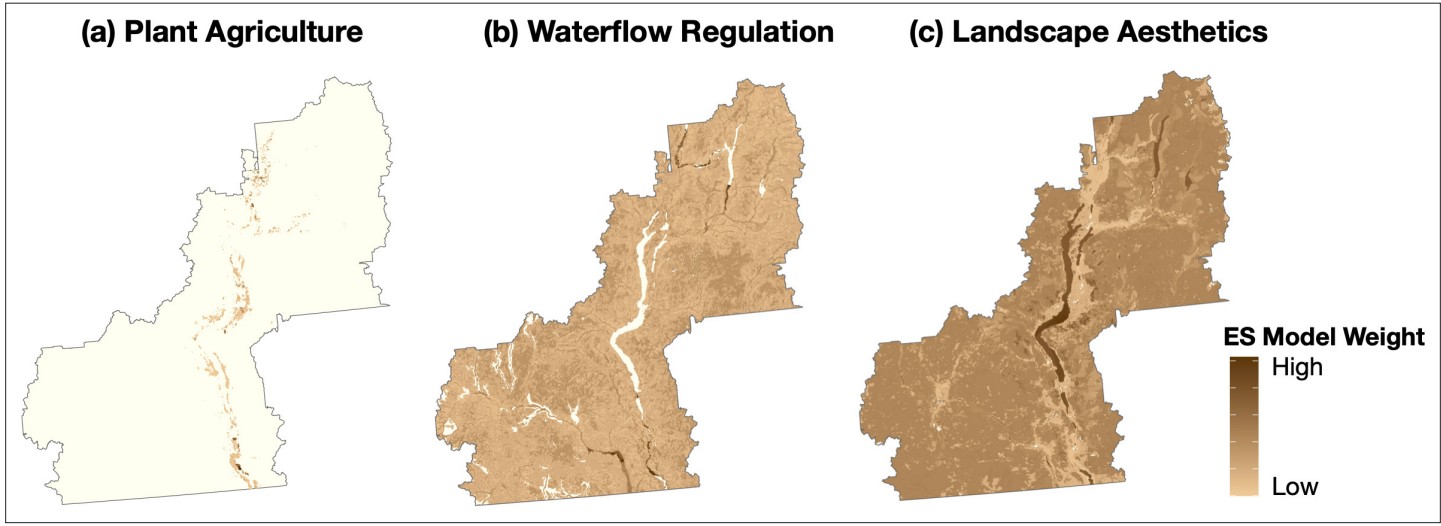

**Landscape aesthetics supply** supports human well-being and is a driver of settlement and tourism in the region. It was modelled based on perceived values of different landcover types, and on the visual accessibility of these terrestrial and aquatic areas.

**Plant agriculture supply** in the region are areas that grow tree fruits and nuts, vines and grapes, cereals, rotation crops, vegetables, berries, and other specialty foods. They are concentrated along the valley bottom.

**Waterflow regulation supply** is provided by terrestrial areas that sustain water delivery in dedicated areas, and that protect against flooding and droughts. It was modelled as a function of soil texture, slope, landcover perviousness, and functionally relevant ecosystem types.

The iconic agricultural landscapes of the Okanagan region contribute to their aesthetics value, which is represented by unidirectional **connections from plant agriculture to landscape aesthetics** where these supply areas **overlap**.

**Waterflow regulation** areas are **connected across** the landscape unidirectionally **to all other ES areas**, including waterflow itself, by maintaining hydrological functions as water moves downslope.

Where supply areas **overlap**, **waterflow regulation** is **connected** bidirectionally to the maintenance of **plant crop** production and of **landscape aesthetics**. In turn, crop and aesthetically-valuable areas help maintain waterflow regulation where they contain supporting ecological characteristics.

**Figure 1.** Schematic and definitions for ES supply areas and functional connections in the case study landscape.

base our approach on existing, and relatively straightforward, spatially co-occurring ES interaction and LCP corridor methods to present a first step toward representing functional connectivity between multiple ES. Our multi-step approach has three specific objectives: (1) to define the ecological process-based connectivity mechanisms between different types of ES supply; (2) to spatially map and quantify these connections while accounting for LULC heterogeneity; (3) to compare coverages of supply areas and functional connections across different types of LULC.

## (a) Plant Agriculture     (b) Waterflow Regulation     (c) Landscape Aesthetics

ES Model Weight
High

Low

**Figure 2.** Maps showing the original, full-extent of distribution and weighting for ES supply areas in the case study landscape, including (**a**) plant food agriculture, (**b**) waterflow regulation, and (**c**) landscape aesthetics (*Field et al., 2017*).

## Results

### Distribution and values of ES supply areas

Based on existing ES supply area mapping from the study area (*Field et al., 2017*; *Figure 2*), top-valued 50% supply areas were distributed north-to-south across our study region (*Figure 3b–d*). Plant foods are grown primarily in valley bottom areas in the Okanagan, and thus PA supply areas (n = 1497) were concentrated in lower-elevation and population-dense regions with similar coverage to the original PA map (distribution of specific crop types detailed in *Field et al., 2017*). The highest-value PA supply areas were coincident with the largest farm parcels, present in the agriculture-rich areas of the south, north and east-central Okanagan. Given the extensive coverage of their original model results, top-value supply areas for both WF (n = 7350) and LA (n = 5262) were distributed fairly evenly across the entire study area. The highest-value WF supplies were associated with stream riparian areas in larger, partially protected sub-basins of the southwest, and with riparian and wetland complexes in the central- and north-east. Our results suggest that the highest-value LA supplies were associated with large areas of upland forests, rivers, lakes, and protected parkland in the southwest and northeast, with relatively lower cumulative LA values in the more heavily-populated valley bottom. It is worth noting that, as our method of delineating distinct LA supply areas was based on the amalgamation of immediately adjacent raster cells, there were several large LA supply areas that may or may not be subjectively interpreted by human consumers as part of a single supply area. Issues with inherent subjectivity around LA mapping and assessment are common (e.g. *van Zanten et al., 2016*, see also *Daniel et al., 2012*), and this could lead to variable results in strength and physical location of cultural supply areas and their inter- and intra-ES linkages. Even nuances within a single cultural ES valuation method can lead to complex results; for example, tourist's aesthetic appreciation of landscape features can differ from that of residents (*Beza, 2010*). That said, the focus of this study is not on how to produce the most accurate spatial representation of ES and their connections, but is on demonstrating a connectivity-based approach for visualizing and evaluating multi-ES relationships. The original LA value distribution map is reproduced in *Figure 2c* (*Field et al., 2017*).

### Distribution and values of functional connections between ES supplies

The spatial distribution and value of connections between overlapping ES were predictable based on the extents of supply area mapping and on the functional theory we applied to link weighting. Bi-directional overlap links between WF and LA (n = 9363 in each direction) were distributed across the entire study are (*Figure 3e, h*). The highest-value links from LA to WF were associated with stream and lake riparian areas in both populated and remote valleys in the north, with riparian and wetland complexes in the central-east, and with remote stream and river riparian areas in the southwest. Similarly, the highest-value links from WF to LA were present in stream and river riparian areas in the southwest, and with stream riparian and wetland complexes in the central-east. For overlapping connections from PA to LA (n = 174), link distribution was sparse throughout the valley bottom and limited to croplands with high aesthetic value; primarily associated with vineyards and orchards (*Figure 3f*). In terms of bi-directional overlap connections, the majority of PA supply areas were connected with WF regulation areas throughout the valley bottoms (WF to PA n = 1,220; PA to WF n = 1320), with highly-weighted links typically associated with cultivated lands, fields, crop transitions, vineyards, and orchards near to (or containing) riparian, floodplain, and/or wetland areas (*Figure 3g and i*).

Topographic links from high-value WF supplies to other ES supply areas revealed corridors variable in length and weight flowing across the landscape, sometimes linking ES supplies ~ 200 kms apart. Between pairs of spatially isolated WF areas, corridors (n = 484,602) approximated the location of watercourses (*FLNRO et al., 2017*), as was expected due to the elevation-based LCP resistance surface used to simulate surface waterflow. The highest-value WF-WF corridors were observed through the large central Okanagan Lake system and several of its relatively low-order tributaries; in high-order valley-bottom rivers, streams and lakes in the southwest; and in the larger valley-bottom rivers of the northeast. These observations resulted from connections between WF supply areas and the influential landscape features (ILFs = floodplains, riparian areas, wetlands, seasonally flooded fields; *Field et al., 2017*) that are prevalent next to valley-bottom aquatic areas. Flowing from WF to LA supply areas, corridors were scattered throughout the study area (n = 2864), with the majority of links associated with the more populated valley-bottom areas in the central Okanagan basin, and with the highest-value links in higher-order streams where sub-basins contained larger numbers of WF

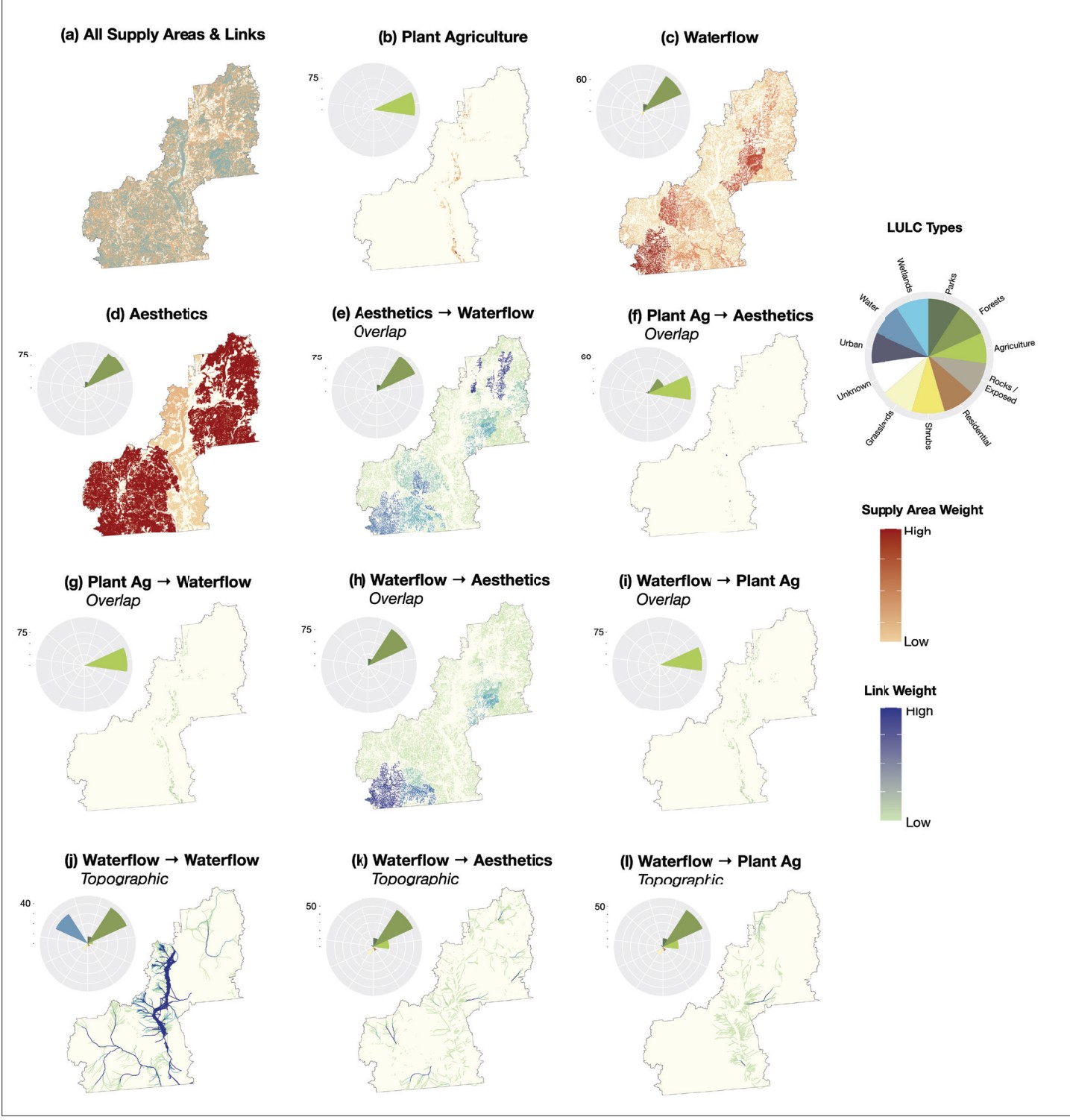

**Figure 3.** Distribution and weighting of top-50%-valued ES supply areas and functional connections on the case study landscape. Insets show (**a**) all top-value supply areas and links; top-value supply areas for (**b**) plant food agriculture (PA), (**c**) waterflow regulation (WF), and (**d**) landscape aesthetics (LA); overlapping connections from (**e**) LA to WF, (**f**) PA to LA, (**g**) PA to WF, (**h**) WF to LA, and (**i**) WF to PA; and topographic connections from (**j**) WF to WF, (**k**) WF to PA, and (**l**) WF to LA. Adjacent circular coxcomb charts represent the proportion of ES supply and link areas covered by major LULC types. LULC types are color-coded and include forests, agriculture, rocks/exposed areas, residential areas, shrubs, grasslands, urban areas, water, wetlands, and areas with unknown use and/or cover.

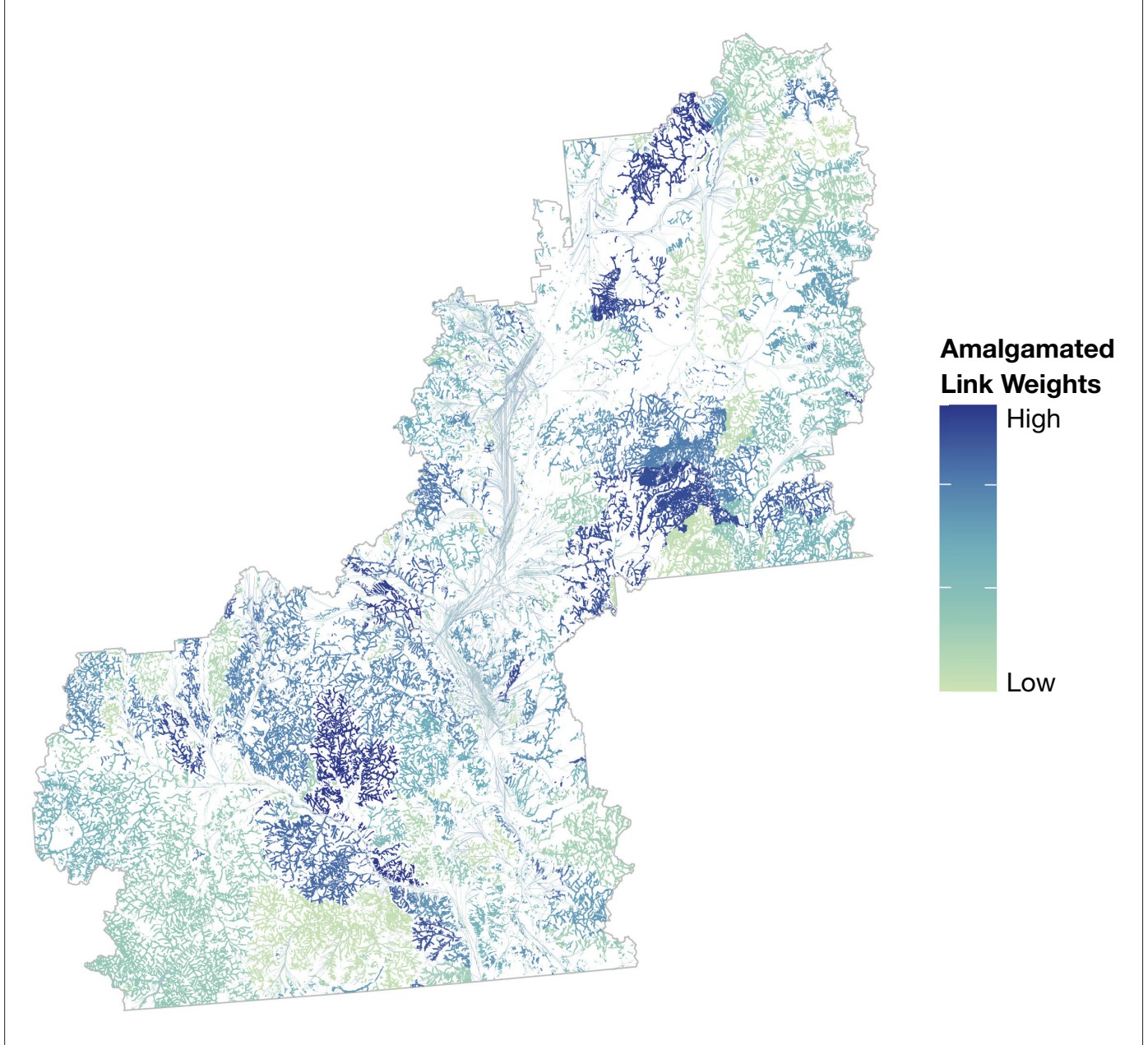

**Figure 4.** Distribution and weighting of link values amalgamated across all eight (8) overlapping and topographic link types across the case study landscape.

supply areas upstream of one or several LA supply areas. Connections to PA were only possible where farmlands were present within the sub-basin of the associated WF supply area, or downstream where farms were within ILF zones. Therefore, such corridors were concentrated in sub-basins along the central valley-bottom (n = 5,256), with particularly high weights in a northern agricultural valley, in the largest sub-basin in the central Okanagan watershed, and in one southern basin (*Figure 3I*). A general trend we observed for all topographic links was the co-occurrence of higher value corridors with larger rivers and streams, rather than being associated with smaller headwater streams. This was a result of the culmination of overlapping corridors from several headwater WF areas in the lower elevation stream valleys that had the largest number of supply areas for the related ES pair type.

When all link types were included on a map of accumulated weights, it highlighted expansive networks of high-value functional connectivity corridors between all three ES types and distributed

across the entire landscape (*Figure 4*). The highest-value link areas were found in low- and mid-elevation riparian areas across the landscape; in a mid-elevation wetland complex of the eastern-central region; in riparian and surface waterflow corridors associated with a large eastern-central sub-basin; and generally, in areas where several (or all) of the eight link types co-occurred. Notably, the accumulation map revealed that several of the highest-value areas were not coincident with the highest-value on any of the individual link-type maps (*Figure 3e–l*), and were sometimes in relatively remote, higher elevation areas.

## Spatial coverage of supply area and linkages across LULC types

The distribution of ES supply areas and links were only in part determined by the underlying LULC types included in the original model parameters defined by *Field et al., 2017*. In decreasing order of coverage of our study area (21,580 km$^2$), the high-level LULC types are as follows: 76.7% forest; 11.9% park (NB: overlaps with forest, grassland, shrub, rock and exposed categories); 6.8% grassland; 5.8% shrub; 3.7% agriculture; 3.0% waterbodies (lakes, rivers, reservoirs); 1.6% residential; 1.3% rock and exposed land; 0.8% wetland; 0.3% urban; and 0.1% unknown (Appendix 2). For the subset top-valued ES areas, almost all PA supplies were, unsurprisingly, on agricultural lands (~100.0%), but only covered 15.8% of all croplands in the region. Both WF and LA supplies were found mainly on forested lands (78.7% and 93.0%, respectively) and within parks (14.0% and 13.1%, respectively). Distribution of top-valued WF supplies covered large portions of most LULC types in the study area (19.7–50.6%; NB: 0% aquatic), including 99.6% of all mapped wetlands. Top-valued LA supplies spanned the majority of aquatic (98.8%), forested (81.9%), park (74.4%), and wetland (69.1%) LULC types (*Figure 3a–d*).

Similar trends in LULC coverage were observed for overlapping connections, with links from PA to LA found mainly on agricultural lands (69.7%) and in forests (28.5%). In both directions between PA and WF, connections were mainly on agricultural lands (both 98.0%), and covered 5.8% and 5.7% of all croplands in our study area from PA to WF and from WF to PA, respectively. In both directions between WF and LA, connections were mainly on forested land (90.8% and 91.1%, respectively) and in parks (14.7% and 14.6%, respectively). Moreover, these links covered large portions of all wetlands, parks and forests (from WF to LA 68.0%, 39.2% and 37.6%; from WF to LA 68.0%, 39.2% and 37.8%, respectively) in the study region (*Figure 3e-i*).

For topographic corridors, we found that LULC coverage was less consistent with relevant ES supply area coverages. Corridors between different WF supply areas were found mainly in forested (42.9%) and aquatic (35.2%) areas, with more minor distribution in park (7.1%), grassland (6.9%), and agricultural (5.2%) LULC types. Notably, topographic WF corridors covered 24.3% of the entire aquatic areas found in our study region. From WF to PA, corridors mainly traversed forested areas (54.1%), followed by agricultural (19.8%), park (10.7%), grassland (10.6%), and residential (5.4%) LULC types. From WF to LA, corridors were mainly found in forested areas (57.3%), followed by parks (14.5%), agriculture (12.7%), and grasslands (10.2%; *Figure 3j-l*). All LULC overlay analyses results are summarized in Appendix 3.

## Discussion

We applied a novel approach to mapping and modeling the functional connectivity between multiple types of ES across a regional landscape. By testing the application of this approach on three ES categories for a case study area, we identified and mapped eight link types connecting ES supply areas on the landscape. The results demonstrate the variety of ways categorically divergent ES can exhibit interdependencies related to their production potential, and the importance of considering these dependencies in land use planning for ecological connectivity.

### The case study: ES connectivity across a heterogeneous regional landscape

The connections we observed between ES revealed high-value multifunctional linkages on the landscape that were not necessarily predictable from supply area mapping. Across all link types we found heterogeneous distribution as well as spatially distinct areas of markedly higher value, or 'hotspots' of connectivity, relative to surrounding areas (e.g. *Alemu et al., 2021*). But one surprising observation is that the weighted amalgamation of all eight link types uncovered areas of high-value connectivity that

were not present on any of the ES supply or pairwise link maps. This finding points to nuances that can be discovered when multiple ES and multiple linkage types are assessed together, and suggests that the spatial focus of planning for optimal service provisioning may shift when functional relationships between several ES are considered. Ultimately, such multifunctional areas represent possible conservation priorities that, if degraded or lost, may cause significant disruption of ES connectivity networks. Understanding the complexity of interactions between multiple ES has been highlighted as a critical challenge in planning for sustainable multifunctional landscapes in the face of changing environmental conditions and management interventions (*Dee et al., 2017*). A recent review of studies that have modeled interactions between multiple ES found that a large proportion did so from the perspective of co-occurring production synergies and trade-offs, but that the examination of flows, and the identification and quantification of explicit functional relationships remain largely unexplored (*Agudelo et al., 2020*). Ultimately, simultaneously modeling multiple ES continues to be difficult in part because of methodological inadequacies and the complexity of the ecological processes involved (*Kolosz et al., 2018*). Our approach provides a new and flexible framework that can help address these challenges.

From initiation points within WF supply areas, our modeling revealed several functional connections that operate over both short and long distances. Some of these topographic corridors extended over 200 km within the boundaries of our study area and, based on the underlying ecological process theory, also extend across the Canada-USA border to wetlands, riparian areas, seasonally flooded agricultural fields, and service supply areas along the extent of the Columbia River to the Pacific coast confluence between Washington and Oregon states, more than 1100 km downstream from the originating supply areas in our study region. Similar long-range connectivity may be observed for other water-related ES (e.g. water provisioning, water quality), as both mean-annual water volume and water quality have been found to be heavily influenced by first-order headwater catchments, even in watersheds with large high-order rivers (*Alexander et al., 2007*; *Freeman et al., 2007*). Additionally, WF exhibits close- and long-range interactions with many other ES not modeled in our study. For example, water extraction and damming to take advantage of freshwater provisioning supplies alters natural hydrological regimes (e.g. *Jackson et al., 2001*); afforestation reduces peak and maintains base flows (e.g. *Zhang et al., 2007*; *Power, 2010*), whereas deforestation destabilizes flows (*Mäler et al., 2013*); areas providing WF supply help to decrease pollution, flood-related turbidity, and residence time of chemicals in lakes (*Burmil et al., 1999*; *Blackstock, 2001*; *Jackson et al., 2001*; *Bennett et al., 2009*); certain pollination services can be facilitated by moving water (*Biesmeijer et al., 2006*); and some recreational activities are dependent on the maintenance of waterflow (e.g. fishing, kayaking; *Burmil et al., 1999*). Based on our observations of the potential for both short- and long-range functional connectivity, ES planning for other water-related services should also consider the potential impacts of management interventions on related services areas and management jurisdictions downstream.

Our study demonstrates that functional connections between ES often span several LULC categories, and that trends in dominant cover types may be unexpected relative to those associated with related supply areas. Certain areas or cover types are sometimes considered 'hotspots' for ES production, that is, they provide several different, often high-value, ES (e.g. *Qiu and Turner, 2013*). For example, wetlands provide flood and flow control, storm protection, erosion control, groundwater supply, water quality maintenance, nutrient waste disposal, habitat to support fishing and hunting, natural materials, biodiversity, micro-climate stabilization, carbon sequestration, recreation, and aesthetic value (*Brander et al., 2006*). Agricultural lands can provide many ES beyond food for humans, such as habitat and food for pollinators, biological pest control (e.g. *Loos et al., 2019*), and tourism (e.g. *Wagner and White, 2009*). We assessed the potential for LULC-associated connectivity hotspots in our region using LULC comparisons. Forested lands clearly stand out as being important for the regulating and cultural ES we investigated. Forests are often identified as hubs for maintaining regulating and cultural ES, including surface water provisioning and quality, soil retention, nutrient retention, pollination, carbon storage, climate regulation, habitat quality, and recreation (*Matson et al., 1997*; *Brauman et al., 2007*; *Qiu and Turner, 2013*; *Karimi et al., 2021*). Notably, although parks make up only 11.9% of the study area, they represent important landscapes for WF and LA supply and overlapping connectivity, and as flow corridors between all ES types we investigated. Both the above observations are likely driven by the suite of ecological processes present in complex

forest, grassland and shrub ecosystems (e.g. vegetation-mediated infiltration, *Mills et al., 2004*), and by the contribution of wildlands and parks to LA (*Thompson and Center for Environmental Philosophy, The University of North Texas, 1995*). From the perspective of functional connectivity, our study suggests a need to expand upon the ES 'hotspot' notion by considering that other LULC types beyond those associated with supply areas may be serving as critical corridors for interdependent ES. A clearly delineated example of this is the ecological process links between terrestrial and aquatic ecosystems. Areas of land adjacent to waterbodies are known to provide various regulation services in addition to WF, including erosion and water quality regulation through soil- and vegetation-mediated retention and filtration (*Mills et al., 2004*). Whereas the model parameters we applied for WF preclude supply area coverage within any aquatic areas, the LULC proportions we observed within upland and downslope WF corridors traversed 24.4% of all surface waterbodies in the region and demonstrated that aquatic areas represent some of the most high-value linkages between different production areas for this ES. In addition, croplands proportionally represent the third-largest cover type in the corridors between upland WF and downslope LA supplies, with the majority of these corridors found in riparian zones, or on farms adjacent to wetlands and waterbodies. The synergistic association of WF and PA supply areas has been observed in other ES interaction studies (e.g. *Qiu and Turner, 2013*), and stems from crops', especially deep-rooted perennials, ability to provide a variety of hydrological benefits including increased water infiltration and recharge, reduced runoff, and mitigation of peak flows (*Dabney, 1998*; *Tilman et al., 2002*; *Brauman et al., 2007*; *Power, 2010*). These observations have implications for ecosystem- and habitat-based management programs as LULC types are often imposed as boundaries for interventions and/or institutions (e.g. BC Ministry of Agriculture). Especially in heterogeneous regional landscapes, our results point to potential for increased need for cross-jurisdictional collaboration when planning for functional connectivity in the optimization of multiple ES.

## The conceptual shift: from correlative interactions to functional connectivity

Our approach reveals cross-landscape connectivity processes that represent important drivers of ES production that are undetectable with traditional methods for identifying ES synergies and trade-offs (e.g. *Qiu and Turner, 2013*; *Su and Fu, 2013*; *Tomscha and Gergel, 2016*). It can be used to represent several different types of functional connections, for example, between different ES that occupy the same space, and abiotic movement from one ES supply area to another across the landscape. Identification of links between spatially co-occurring supply areas is similar to a representation of paired ES 'interactions', a concept for identifying synergies and trade-offs among services, as well as identifying groups of services that repeatedly occur together across a landscape (i.e. 'ES bundles'; *Bennett et al., 2009*). Our methods take a closer look at the concept of ES interactions by explicitly representing the mechanisms behind the co-occurrence of ES in the delineation and valuation of these areas (*Bennett et al., 2009*). Investigation of interaction mechanisms with respect to multi-ES assessment has been highlighted as a crucial step toward providing more rigorous information to inform the management of multifunctional landscapes (*Alemu et al., 2021*; *Thierry et al., 2021*), and our study is one of the few to provide this information at the regional scale (*Agudelo et al., 2020*).

Areas of ES supply are not necessarily spatially congruent with the discrete structural components traditionally considered in landscape connectivity frameworks (e.g. habitat patches, specific LULC types); therefore, linkages between ES are also unlikely to be coincident with these components (e.g. movement of organisms and matter; *Brooks, 2003*). For example, the global benefit of carbon storage and sequestration depends only on the quantity of natural landcover, and not the spatial arrangement of patches (*Mitchell et al., 2015*). Although protected areas and intact habitat patches are important spaces for some of the plants, animals, and abiotic ecosystem components responsible for providing ES, provisioning and flows are not bound by human-defined reserve areas, and many ES are produced completely by and interact with one-another in human-modified landscapes (*Schröter et al., 2019*). Further, connectivity of certain ES will be limited by distance thresholds and/or human or ecological barriers to the flow of ecological processes. For example, crop production can benefit from interspersion of natural habitat throughout agricultural landscapes, which can increase pollination and pest control services delivery from species that can only move limited distances from their habitat patches (*Tscharntke et al., 2005*). There may be spatial congruency between existing

**Table 1.** Rationale behind functional connection mechanisms, directionality, and weighting between top-value ES supply areas. Directionality is represented by the top row as originating (or 'source') ES supply areas; and the left column as recipient (or 'sink') ES supply areas (PA, WF, or LA). Functional connection mechanisms are distinguished by Link Type (i.e., Overlap or Topographic), and their definitions and weighting rationale are provided in matrix cells. If no functional connection exists from one ES to another, the matrix cell indicates that there is 'None' in either direction, or that there is a connection in the '*Other direction*'. Superscripts denote the following references: (1) *Crossman et al., 2013*; (2) *Power, 2010*; (3) *Daniel et al., 2012*; (4) *Zhang et al., 2007*; (5) *Bennett et al., 2009*; (6) *Poff et al., 1997*; (7) *Burmil et al., 1999*; (8) *Raudsepp-Hearne et al., 2010*; (9) *Nicholls and Altieri, 2012*.

| | linked from →<br>linked to ↓ | Link Type | **Supply areas** | | |
|---|---|---|---|---|---|
| | | | Plant Agriculture (PA)<br>*supply area weight*: potential PA crop area (ha) | Waterflow Regulation (WF)<br>*supply area weight*: summed WF model value | Landscape Aesthetics (LA)<br>*supply area weight*: area (ha) x LA model value |
| Supply Areas | Plant Agriculture | Overlap | None | WF regulation on PA croplands[4,5]<br>*link weight*: summed WF model supply area values within PA supply area | *Other direction* |
| | | Topographic | | WF regulation downslope[4,5]<br>*link weight*: summed WF model values along LCP pathway from WF to PA supply area | None |
| | Waterflow Regulation | Overlap | PA croplands providing WF regulation[1,2]<br>*link weight*: all summed WF model values within entire PA supply area | None | LA areas providing WF regulation[8,9]<br>*link weight*: summed WF model values within LA supply area |
| | | Topographic | *Other direction* | WF regulation downslope[6]<br>*link weight*: summed WF model values along LCP pathway from WF$_1$ to WF$_2$ supply area | *Other direction* |
| | Landscape Aesthetics | Overlap | PA cropland providing LA[3]<br>*link weight*: summed LA model values within PA supply area | WF regulation on LA areas[7]<br>*link weight*: summed WF model supply area values within LA supply area | None |
| | | Topographic | None | WF regulation downslope[7]<br>*link weight*: summed WF model values along LCP pathway from WF to LA supply area | |

wildlife movement corridors and certain regulating services, especially those that depend on the movement of organisms for their delivery (e.g. pollinators, disease control, pests and their predators, seed dispersal; *Kremen et al., 2007*), which suggests that there may be opportunities for win-win conservation initiatives for wildlife and ES together. Our approach can be used to explore this possibility, and to explicitly map and assesses the mechanisms behind distance-threshold-mediated and cross-landscape ES interactions in general.

## Limitations and opportunities for future work

We identify several limitations of our approach, and suggest related opportunities for future work. Firstly, we only included three ES in our study and comparisons, a decision driven by available data and desire to clearly test a new approach while using a diverse subset of ES. A small number of tested ES means we are limited in the generalizations we can make, especially as they pertain to LULC- and ecosystem-relevance of the potential for connectivity 'hotspots'. Investigating a limited number of ES is common among studies that model interactions among ES (*Agudelo et al., 2020*), with data limitations, complexity of socio-ecological processes involved, and methodological gaps cited as barriers to inclusion of all ES (*Kolosz et al., 2018*). However, our choice to test only three ES was motivated by our goal to provide a straightforward case study of how each of the three broad ES categories (i.e. non 'supporting'; *Millennium Ecosystem Assessment, 2005*) can be represented in the same study. Our approach is easily adaptable to including an unlimited number of ES, though the complexity in representing the functional connections between them may increase disproportionately to the number of ES included, and limited data and/or gaps in our understanding of interaction mechanisms may preclude modelling of certain pairwise relationships (*Field and Parrott, 2017*). As evidenced by the lack of known functional links between some of the ES in our study (e.g. topographic links between PA and LA; *Table 1*), some pairs of ES may not exhibit inter- or intra-ES connectivity, although these can still be included on maps as disconnected supply areas on the landscape.

Further, we note that only synergistic interactions were identified among the case study ES we included, but no trade-offs were represented. Although the latter has been identified among the ES tested in our study (e.g. water extraction for agriculture can disrupt hydrologic cycles; *Janssen et al., 2006*), we did not model these due to lack of data on specific trade-off mechanisms. The incorporation of other services may reveal both positive and negative effects of connectivity on ES supply as a result of complex multi-ES interactions; for example, the rate of waterflow through riparian areas may increase filtration and water quality regulation, but decrease downstream water provisioning (*Mitchell et al., 2013*). The presence of potential trade-offs, as well as ecosystem dis-services (e.g. competition for water and pollination among different LULC types; spread of pests and diseases; *Zhang et al., 2007*), is of critical importance to informing management, as the optimization of all ES on a landscape is usually not simultaneously possible (e.g. *Qiu and Turner, 2013*). We encourage future applications of our approach to represent trade-offs and negatively-valued functional connections between ES where appropriate.

Our case study maps and quantifies relationships between ES at a snapshot in time due to lack of temporal data available for the ES tested in our study. However, modifications of natural landcover can change the number, size, shape, isolation, and distribution of ecological patches across the landscape and their proximity to human beneficiaries, all of which may lead to positive, negative or neutral impacts on ES supply and flow (*Mitchell et al., 2015*). For example, in the face of climate change increases in dryland aridity causes grasslands and savannahs to metamorphose into shrublands as the latter grow better in sandy, nutrient-poor soils (*D'Odorico et al., 2012*; *Phillips et al., 2019*). In our study area specifically, such a shift would have implications for ES coverage and value through the dependency of model variables on underlying vegetation characteristics (*Field et al., 2017*), and thus influence future ES production and connectivity. The ultimate impacts of landscape changes on ES are dependent on the structure and quantity of such changes, and on the biophysical process, ecosystem functions, species, and human activities driving the ES supply of interest, as well as the flows to and demands of human beneficiaries (*Mitchell et al., 2015*). Further, it has been shown that spatial correlations between pairs of ES can exhibit inter-annual variability (e.g. *Renard et al., 2015*; *Li et al., 2017*), and that snapshots in time are not good predictors of how their relationships may change over time (e.g *Mitchell et al., 2020*). Future studies could use the ES connectivity framework presented here to assess how changes in LULC ultimately have cascading impacts on multiple ES across a landscape (*Bagstad et al., 2013*; *Grêt-Regamey et al., 2017*; *Rieb et al., 2017*), which can practically be achieved by incorporating seasonal and inter-annual variations in ES supply, demand and functional connectivity (e.g. increases in fresh water provisioning during dry months; *Field and Parrott, 2017*).

We acknowledge that that other existing methods for spatially identifying and evaluating spatial connectivity may be more appropriate for certain relationships between other ES not tested in this study. Examples include least cost corridors (*Singleton, 2002*), circuit theory (*McRae and Beier, 2007*; *McRae et al., 2008*), graph theory (*Fall et al., 2007*; *Pinto and Keitt, 2008*, *Rayfield et al., 2011*), spatial networks (*Phillips et al., 2008*; *Parks et al., 2013*), Euclidean distance mapping (*Doak et al., 1992*), radius buffers (*Laliberté and St-Laurent, 2020*), and deterministic eight models (*Mark, 1984*). These could also be combined with dynamic modeling approaches (e.g. Bayesian belief networks; scenario modeling) that can incorporate measures of uncertainty (e.g. *Karimi et al., 2021*; *Sahraoui et al., 2021*; Appendix 4). Specifically, incorporating measurable changes in ES characteristics (e.g. supply quantities or distributions) will allow researchers to test the degree of influence one ES supply area has on another. We chose to employ only LCP analysis mainly because the topographic ES flows in our study all originated at WF supply areas, corridors all were to represent the ecological process of water flowing downslope, and because LCP has been shown to be a valid method for approximating drainage networks while being capable of overcoming issues around topographic depressions (*Melles et al., 2011*). Therefore, a DEM-driven model representing water moving downslope was deemed the most appropriate for these types of ES connections in our study region, which was supported by validation analyses (Appendix 5), while providing relatively simple and accessible representations of corridors between supply areas to support the primary goal of this paper, that is, to demonstrate a novel approach for conceptualizing how the provisioning of ES are functionally connected across a landscape. Future research should compare and validate alternative spatial connectivity mapping and valuing approaches (e.g. *Melles et al., 2011*) for predicting process-based movement between other ES types, with validation approaches tailored to the specific ES under study (e.g. Appendix 5).

For example, connections between pollination supply areas and PA could be represented by first identifying pollinator habitat, which may encompass natural and semi-natural habitat areas within and/or adjacent to PA areas (e.g. *Power, 2010*). Then the functional link mechanism may be represented by modeling pollinator movement between habitat areas (e.g. three-dimensional surface models; *Abdel Moniem and Holland, 2013*). As long as the researcher clearly defines the known functional mechanisms a priori and selects the appropriate model(s), our approach is flexible in that it allows for a variety of functional connections to be spatially represented, and for the use of several methods in delineating functional relationships between multiple ES which can then be incorporated into the same map for spatial representation of such connections. Within the defined functional mechanisms, models can allow for the incorporation of complex interactions between ES, such as nonlinear relationships and threshold effects (*Thierry et al., 2021*). We note that certain multi-ES relationships may not be amenable to spatial representation or assessment using connectivity mapping (e.g. air quality with erosion control; animal agriculture with PA from the perspective of direct, non-fodder sources of food for humans, though known fertilizer contributions may allow for a directional functional connection to be defined), but their supply areas can still be included on maps to display distribution, values, or other metrics.

## Conclusions

Our study provides a new approach for the assessment of multiple ES and provides important information on the spatial interconnectivity of a variety of divergent types of ES across a diverse temperate landscape in southern interior British Columbia. We are confident that providing a tool for visualization of multiple ES will help address several ongoing challenges: increase awareness and understanding of how dependent humans are on nature; highlight a need to maintain landscape connectivity to support ecological functioning; advance the interdisciplinary science around the ES concept; and help move toward incorporating this science into management of natural capital (*Guerry et al., 2015*). As the ES concept continues to be developed and refined, considering how ES operate within the context of interconnected, complex social-ecological systems will help improve our ability to meaningfully incorporate multiple ES into decision-making and planning at the landscape scale. Overall, our methods not only allow for the explicit incorporation of the current knowledge of the ecological processes driving linkages between multiple ES, but they also provide decision makers mapping tools that show where these connections occur on the landscape and how valuable they are to ES flows and production potential. Thus, our approach can help guide planners in predicting how intervention(s) in specific location(s) are likely to have synergistic or antagonistic impacts on ES supply areas in other, sometimes distant places.

# Materials and methods

**Key resources table**

| Reagent type (species) or resource | Designation | Source or reference | Identifiers | Additional information |
|---|---|---|---|---|
| Software, algorithm | R (v.3.6.2) | *R Development Core Team, 2013* | RRID:SCR_001905 | |
| Software, algorithm | ArcGIS (v.10.7.1) | *ESRI, 2011* | RRID:SCR_011081 | |

Our case study area spans the Okanagan region in British Columbia (BC), Canada, which we use to demonstrate a multi-ES connectivity mapping approach for informing landscape planning (*Figure 5*). It is located in the south-central interior of BC, is a biodiversity hotspot within Canada and one of North America's most endangered semi-arid ecoregions (*Warman et al., 2004*; *Kerr and Cihlar, 2004*), has a highly diverse assemblage of land use types (see *Caslys Consulting Ltd, 2013*; Appendix 2), and covers 21,580 km$^2$ from ~276 to 2774 masl. The diversity of LULC and ecosystem types of this multifunctional landscape allow the results to be more widely applicable to other regions relative to a study of a more homogenous landscape.

## Approach

We developed and tested a flexible approach that can be used to map the functional relationships between multiple ES on a landscape. It is flexible in that it can be adapted to various decision and

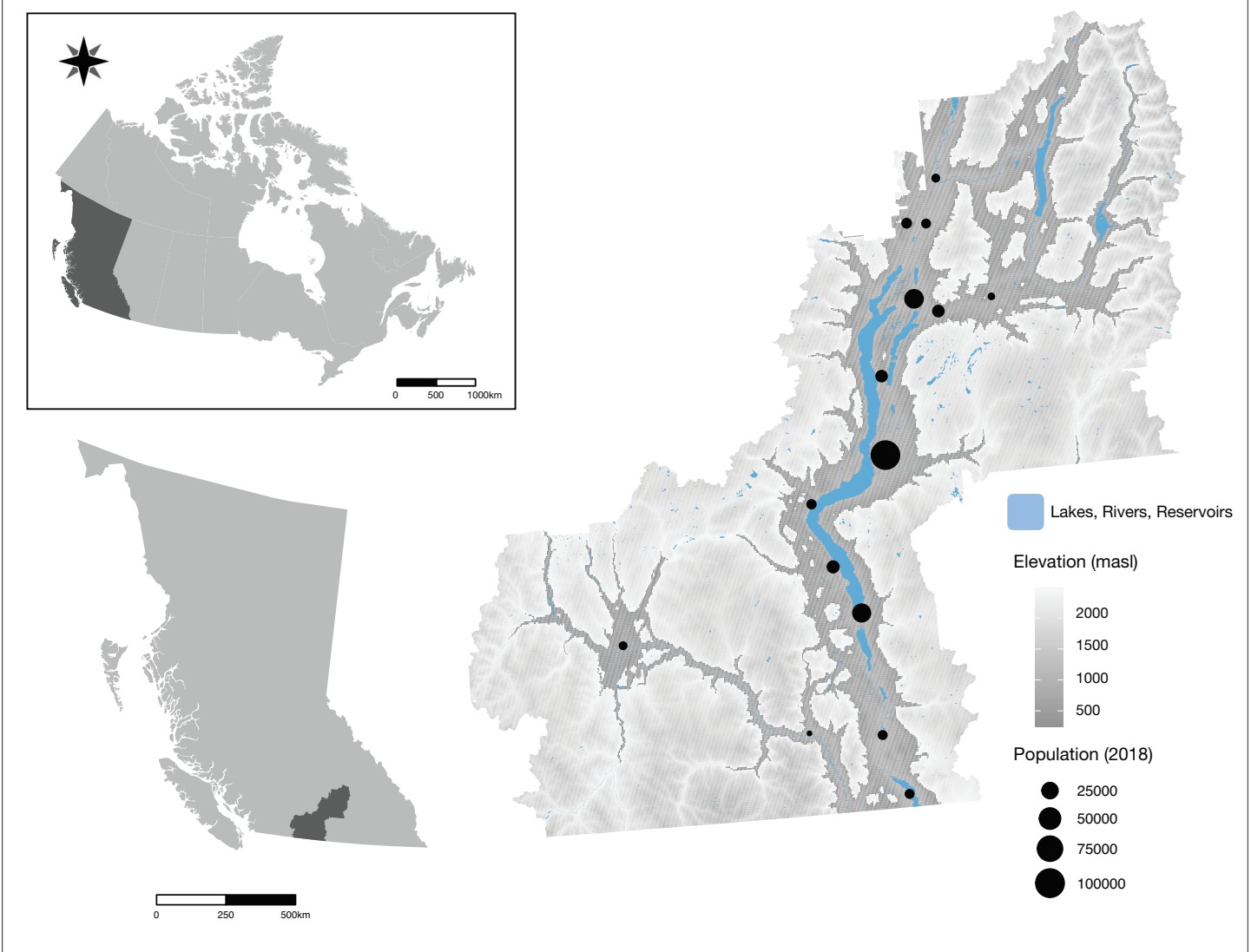

**Figure 5.** Location of the case study landscape in southern interior 'Okanagan' region of British Columbia, Canada. Major waterbodies, elevation (masl), and the most populous cities and towns in Okanagan regional districts are indicated.

research contexts (e.g. *Value of Nature to Canadians Study Taskforce, 2017*), and can incorporate a variety of methods and models for mapping ES provisioning and flows. *Figure 6* provides high-level guidelines for researchers with the intent of supporting standards for comprehensive ES assessments (e.g. *Crossman et al., 2013*). The guidelines focus on the general technical approach for producing spatial assessment tools that will inform goals of researchers and/or decision-makers; other aspects required to produce thorough ES assessments (e.g. defining issue and context; time and expertise resource logistics; communicating results) are presented elsewhere (e.g. *Value of Nature to Canadians Study Taskforce, 2017*).

## Original ES supply data

We obtained existing data on the spatial distribution of the 'supply' of three ES: (1) plant food agriculture ('PA' herein; provisioning = products obtained from ecosystems); (2) waterflow regulation ('WF'; regulating = abiotic and biotic processes that moderate natural phenomena); and landscape aesthetics ('LA'; cultural = non-material characteristics that benefit human well-being; *Millennium Ecosystem Assessment, 2005*) produced for our study area by *Field et al., 2017* (*Figure 2*; *Field, 2021*). *Field et al., 2017* mapped *Field et al., 2017* ecosystem attributes and quantified their potential contribution to ES supply based on environmental characteristics and functions that are known to be related

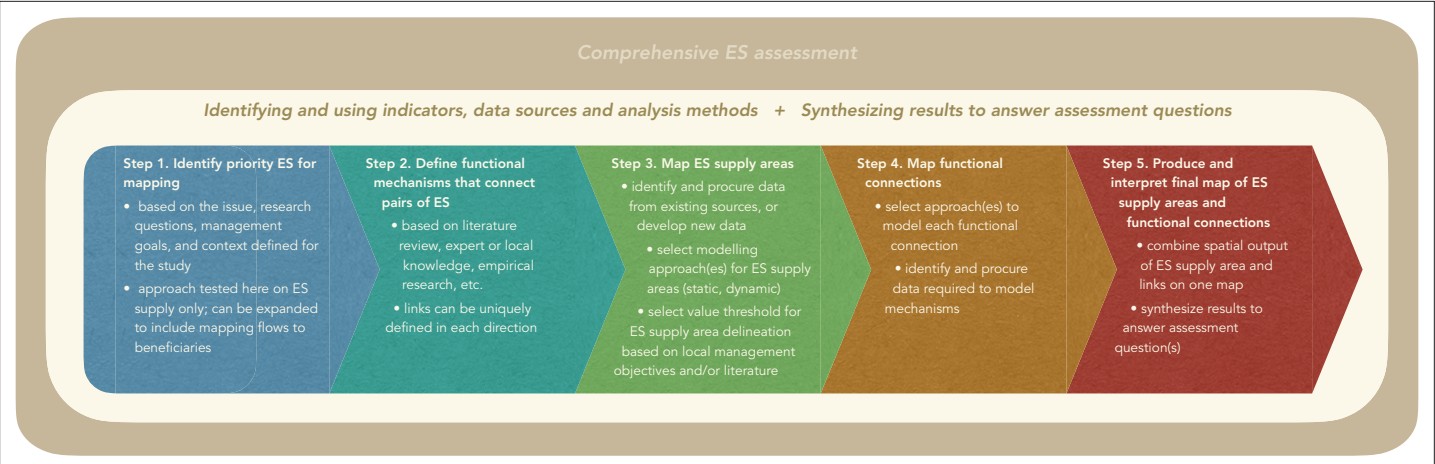

**Figure 6.** General approach guidelines for mapping the functional relationships between multiples ES within a comprehensive ES assessment framework (e.g. *Value of Nature to Canadians Study Taskforce, 2017*).

to ES production; on explicit incorporation of perceived benefits to humans; or a combination of the two methods (*Jakeman and Letcher, 2003*; *Vigerstol and Aukema, 2011*; *Field et al., 2017*). Spatial data sources for original maps included LULC indicators, remote sensing image interpretation, and were supported by some field-validations. For analytical consistency, raster data for original mapping were assigned the identical spatial projection by resampling to ~29 m x 29 m resolution based on the size of digital elevation model (DEM) cells; therefore, ES models accounted for fine-scale heterogeneity of parameters across the landscape. Original maps were created using ArcMap 10.2 and 10.4 (*ESRI, 2011*), and R (*R Development Core Team, 2013*). For full details see *Field et al., 2017*; data sources were all updated since 2001, and are summarized in *Appendix 3—table 1*; data are available on the Open Science Framework (OSF; *Field, 2021*).

PA is an economically and culturally important ES in our study area (e.g., *OVEDS, 2013*; *Kyle, 2018*), and its supply was mapped based on the spatial extent of all crop types used directly for human nutrition, all of which are concentrated primarily in valley-bottom areas (*Field et al., 2017*). From these data, we dissolved boundaries between adjacent Agricultural Land Use Inventory (ALUI) polygons which, in some cases, resulted in different crop types being merged into a single node (*MoAg BC Ministry of Agriculture, 2017*). We did this to generalize the mapping of connectivity between PA as a whole, and the other ES considered in this study, as the rationale for the mechanistic connections between PA and the other ES were consistent for all crop types. This outcome fits with our method of PA supply area valuation, which is based solely on potential crop area (ha) and not on crop type.

*Field et al., 2017* mapped the terrestrial areas that provide WF mapped as a function of soil texture, slope, land use and land cover (LULC)-specific perviousness (normalized difference vegetation index, or NDVI; *Appendix 3—table 2*), and functionally relevant ecosystem types including floodplains, riparian areas, wetlands, and seasonally flooded fields ('influential landscape features' -– 'ILF' herein). WF supply areas were defined as those that sustain water delivery in dedicated areas, and protect against flooding and droughts, both of which are persistent environmental concerns in the study region (*Haughian et al., 2012*). Several wetland areas were excluded from the *Field et al., 2017* WF map due to the absence of soil texture data, which was one of the inputs for the waterflow infiltration model. Because wetlands are so critical to supporting WF and are relevant to connectivity mechanisms with several other ES, we added these areas back to our WF map by re-running the infiltration model under the assumption that all wetland areas without soils data have 100% saturated hydraulic conductivity, and then applying an ILF multiplier per *Field et al., 2017*. These resulting raster values ranged from 19 to 1200 (mean = 237.7; st.dev = 120.1), and wetlands coincident with mapped floodplains provided the highest-value WF supply areas in the region.

Lastly, *Field et al., 2017* mapped LA supply areas were on models of perceived values of different LULC types in the region, on 'visual condition' ranging from preserved to manicured lands, and on the visibility of areas from various viewpoints across the case study region. LA supply areas spanned

both terrestrial and large aquatic (i.e. lakes, rivers, manmade reservoirs) areas. We did not separate adjacent terrestrial from aquatic LA supply areas as 13 LULC (10 terrestrial; three aquatic) values were used as input for original LA mapping, in conjunction with two other valuation methods (i.e. tourism brochure assessment; viewshed analysis), and we aimed to keep supply area delineation method-ologically as consistent as possible across different ES types (e.g. for amalgamation of immediately adjacent supply areas). As LULC data overlapped in certain areas, LULC categories were ranked based on data confidence and relevance to mapped ES (*Appendix 6—table 2*). We note that LULC data-sets were used for mapping all three ES types in our study. We do not believe these interlinkages will impact interpretation of our results as the use of LULC indicators is common and typically the best available proxy data for ES mapping (e.g. *Queiroz et al., 2015*), and spatial overlap is an inherent characteristic of ES (e.g. *Bennett et al., 2009*).

## Mapping and valuing ES supply patches

Based on the original ES mapping data produced by *Field et al., 2017*, we first established and valued *supply* area polygons – defined as spatially identifiable regions of higher-than-average-value supply potential – which serve as source and destination patches in a connectivity network (Appendix 1). We then developed a methodology to establish and value functional linkages, or connectivity, between supply areas. Functional connections were of two broad types: (1) overlapping links, which were areas where the supplies of two different types of ES occur in the same place, and there is an underlying process-based connection between them; and (2) topographic links, which were mapped based on the ecological processes that functionally connect the supplies of two ES areas separated in space. Links could exist in one or both directions, with unique mechanisms operating from one ES to another. Lastly, we compared the coverage of top-value ES supplies and their linkages on the major LULC types found in the region. The details of our approach are provided below.

To spatially partition the landscape into ES supply areas and establish the links between them, we developed a rationale based on interdisciplinary methods for assessing complex and connected natural systems (*Bialonski et al., 2010*). From the perspective of the landscapes' capacity to provide ES, we defined subsystems as discrete areas with the greatest potential for providing ES supply (Appendix 7), while the functional interdependencies between such areas were represented by spatial connections (also referred to herein as 'links' or 'corridors'). We delineated ES supply areas based on approaches used in landscape connectivity and ES mapping studies: spatial polygons that represent high-value supply ES patch boundaries (e.g. *Bangash et al., 2013*); and the aggregation of immedi-ately adjacent clusters of high-value supply spatial grid cells (e.g. *Gardner, 1999*; *Urban et al., 2009*; *Qiu and Turner, 2013*; *Field and Parrott, 2017*). Aggregated areas became supply area polygons, and were valued based on the summed raster values therein, then normalized on a unit-less scale from 1 to 10,000. Any areas either lacking the potential for ES supply, or below a high-value supply threshold (Appendix 7), were represented as the landscape matrix through which ecological process-based connections between supply areas could flow (*Field and Parrott, 2017*). In reality, such spatial interaction networks are dynamic through time (*Boesing et al., 2020*), though here we consider a static snapshot of the present state of ES supply in our study region to clearly illustrate real-world application of a novel approach for mapping the ecological relationships underpinning multiple types of ES supply.

## Establishing functional connections between ES supplies

We define ES connectivity as areas on the landscape where one ES supply area influences the provi-sioning of another via underlying ecological processes. We identified spatial interactions between ES supply areas either as those that are connected through their overlap in space, or those that trans-verse the landscape through the relatively low value (i.e. sub-50% threshold) ES matrix. For these two cases respectively, we applied spatial overlay analysis (e.g. *Qiu and Turner, 2013*), or identified flows using a stepwise procedure involving least-cost path (LCP) analyses akin to those applied in wildlife connectivity studies based on species movement and habitat attributes (*Urban et al., 2009*). Move-ment of organisms and matter across a landscape is often specifically defined in a single direction as a result of biophysical (e.g. waterflow, topography) or biological (e.g. movement from source to destina-tion areas) realities, with multiple link types representing qualitatively unique flows that exist between patches (*Zhang et al., 2007*; *Urban et al., 2009*). That is to say, an area on the landscape producing

multiple ES supply types may have functional links between ES of the same type in different locations, between different ES types in the same location, or with different ES types in different locations.

For the three ES we considered in this study, we characterized eight (8) spatial link types by the directional ecological process-based relationships between high-value ES supply areas. The rationale behind these connectivity mechanisms are summarized in *Table 1*. As connectivity model distribution and valuations were based on the original fine-scale supply area mapping, they also accounted for model parameter heterogeneity across the study area. We identified two high-level types of connections: overlapping (n = 5) and topographic (n = 3). Overlapping links were defined as areas where the supplies of two different types of ES occur in the same location on the landscape and a functional relationship exists between two ES. We used the high-value ES supply area maps (*Figure 3b–d*) to identify areas where each pair of ES overlapped (directionally) based on process theory (*Table 1*) using a GIS clip procedure (see Appendix 7 for step-by-step details; *Field, 2021*). The resulting single-part polygons of overlapping links represented the ecological processes connections between spatially co-occurring ES types (*Figure 3e–i*).

Topographic links were based on the ecological processes that functionally connect the supplies of two spatially separated ES areas across the landscape. Based on the three ES we considered, topographic connections always originate at a WF supply area, and represent the influence of upslope water regulation on the maintenance of the natural hydrologic processes that help support PA (e.g. crop growth and nutrient retention; *DeLaney, 1995*; *Nelson et al., 2009*), WF (e.g. natural baseline flow regulation; *Nelson et al., 2009*), and LA (e.g. maintenance of hydrology-dependent vegetation and aquatic features deemed to have high aesthetic value) supplies in downslope areas. We developed a stepwise procedure to create topographic links between ES supply areas. First, we used least cost path analyses (LCP) to map directional ecological corridors from each WF supply area to sub-basin specific, lowest-elevation outlet ('goal') points based on the assumption of downslope surface waterflow over a DEM surface (e.g. *Melles et al., 2011*). Following this, we segmented resulting LCP lines to produce separate topographic corridors between pairs of supply areas (*Appendix 7—figure 2*). We then identified ILFs as additional WF polygons downstream of each sub-basin in the valley-bottom and associated with wetlands, floodplains, riparian areas, and/or seasonally flooded fields, which are functionally linked to upstream hydrological regulation. We connected ILFs to each upstream sub-basin outlet point, and individually merged these sub-basin lines with the LCPs for each WF supply area within that sub-basin (*Appendix 7—figure 2*). Appendix 7 provides details on the above approach and procedures used to address other analytical nuances; a summary of how the topographic link approach was validated is provided in Appendix 5.

In addition to spatially identifying connections between pairs of ES, we quantified the weight of these connections based on assumptions around the functional relationships between ES (e.g. *Urban et al., 2009*). We based valuations on the original ES provisioning maps, which assigned each raster cell in the map an ES value equivalent to the results of the underlying models (*Field, 2021*; *Field et al., 2017*), and on the assumptions summarized in *Table 1* and discussed in Appendix 7. We acknowledge that alternative ecological process models could be used to produce more nuanced or accurate measures of link weightings (e.g., *Cadotte et al., 2011*). However, we chose to base our link quantification on high-level and readily calculable assumptions in an effort to provide simple, replicable, and easily-communicated metrics to inform applied, often resource-limited, decision-making for corridor, conservation, and protected area placement.

## Comparison with regional LULC

To compare the spatial coverage of supply areas and their linkages, and to aid in our assessment of potential uses of ES connectivity results for on-the-ground planning and management, we calculated the proportion of several high-level LULC categories intersected by each of the high-value supply areas and eight link types identified in the above analyses. We selected several LULC categories to provide both local and regional decision-makers additional information about where on the landscape ES connectivity is distributed, including forests, grasslands, shrubs, parks, aquatic areas, wetlands, rock and exposed land, agriculture, residential, and urban areas. We calculated the total area (ha and %) of LULC types covered by each link type, and the proportions of study area total LULC covered by each link.

ArcMap 10.7.1 (*ESRI, 2011*) and R (version 3.6.2; *R Development Core Team, 2013*) packages sp 1.4–5 (*Pebesma and Bivand, 2005*; *Bivand et al., 2013*), sf 0.9–8 (*Pebesma, 2018*), rgdal 1.5–23 (*Bivand et al., 2015*), raster 3.4–5 (*Hijmans and van Etten, 2012*), rgeos 0.5–5 (*Bivand et al., 2017*), maptools 1.1–1 (*Lewin-Koh et al., 2012*), and stringr 1.4.0 (*Wickham, 2010*) were used to build, assess and visualize the ES connectivity map. LCP analyses and subsequent stepwise link refinement were run using R package gdistance 1.3–6 (*van Etten, 2017*). For transparency and reproducibility, data, R scripts and further details on our methodological procedures are available on the OSF (*Field, 2021*; Appendix 8).

## Acknowledgements

We received constructive feedback from L Hooker, J Janmaat and J Pither. The Regional District of the Central Okanagan and the Okanagan Collaborative Conservation Program provided support and collaboration on this project. Funding for this manuscript was provided by a Discovery Grant from the Natural Sciences and Engineering Research Council of Canada (NSERC) awarded to LP, and by a Canada Graduate Scholarship from NSERC awarded to RDF.

## Additional information

### Funding

| Funder | Grant reference number | Author |
| --- | --- | --- |
| Natural Sciences and Engineering Research Council of Canada | Canada Graduate Scholarship | Rachel D Field |
| Natural Sciences and Engineering Research Council of Canada | Discovery Grant | Lael Parrott |

The funders had no role in study design, data collection and interpretation, or the decision to submit the work for publication.

### Author contributions

Rachel D Field, Conceptualization, Data curation, Formal analysis, Funding acquisition, Investigation, Methodology, Visualization, Writing – original draft, Writing – review and editing; Lael Parrott, Conceptualization, Funding acquisition, Project administration, Resources, Supervision, Writing – review and editing

### Author ORCIDs

Rachel D Field ⓘ http://orcid.org/0000-0001-7726-3085
Lael Parrott ⓘ http://orcid.org/0000-0002-3995-3322

### Decision letter and Author response

Decision letter https://doi.org/10.7554/eLife.69395.sa1
Author response https://doi.org/10.7554/eLife.69395.sa2

## Additional files

### Supplementary files
• Transparent reporting form

### Data availability

Data, R scripts and further details on our methodological procedures are available on the Open Science Framework (OSF; https://doi.org/10.17605/OSF.IO/9S4RM).

The following dataset was generated:

| Author(s) | Year | Dataset title | Dataset URL | Database and Identifier |
|---|---|---|---|---|
| Field R, Parrott L | 2021 | Mapping the functional connectivity of ecosystem services supply across a regional landscape - Data - Field & Parrott 2021 | https://doi.org/10.17605/OSF.IO/9S4RM | Open Science Framework, 10.17605/OSF.IO/9S4RM |

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

## Appendix 1

### Background information

ES interactions: Other work has defined 'interactions' as value-based synergies (increase in supply quality and/or quantity of one ES results in supply increase of another), trade-offs (increase in supply of one ES results in decrease of another; *Bennett et al., 2009*), bundles (groups of ES that co-occur repeatedly across a landscape, typically linked to co-variation in LULC types; *Raudsepp-Hearne et al., 2010*; *Lee and Lautenbach, 2016*), or flows (ES interactions from the perspective of beneficiaries) between ES that occur over the same space and time (*Agudelo et al., 2020*).

ES supply: We note that, although we use the term 'supply' to refer to the portion of the ES provisioning delivery chain on which we are focused, two of the ES we have selected can be conceptualized as spanning both the supply and 'flow' aspects of this chain. ES supply refers to the ecological good(s) and service(s) produced by a natural or man-made area on the landscape (*Potschin, 2016*), whereas ES flow represents human access to ES supplies, that is, the transfer of a good and/or service from a supply to a benefit area or actor for use (*Villamagna et al., 2013*; *Schröter et al., 2018*; *Schirpke et al., 2019*; *Vallecillo et al., 2019*). For agriculture and landscape aesthetics, human action is typically required for these services to actually flow to beneficiaries, for example, produce being shipped to grocers; people venturing into nature to enjoy beautiful viewscapes. However, in the case of aesthetics, the data on which we based our mapping was informed by a viewshed analysis, which spatially quantified the potential for people to actually see areas all across the landscape, thereby incorporating an ES flow component. In the context of preventing or minimizing the impacts of flooding, waterflow regulation is provided (i.e., flows from supply to demand areas) when a supply area limits or delays the flow of water (*Luck et al., 2009*), which is typically a temporal dynamic dependent on seasonal temperature and/or weather patterns. Although the existing ES maps we use in this study are static spatial representations of potential supply areas, in the cases of waterflow regulation and landscape aesthetics, the distribution of potential spatial location of flows will still be captured by this mapping. Note also that ES flows are not equivalent to ES connectivity, the latter of which we are defining by the functional ecological interrelationships between different supply areas.

## Appendix 2

### Distribution of major LULC types across the case study region

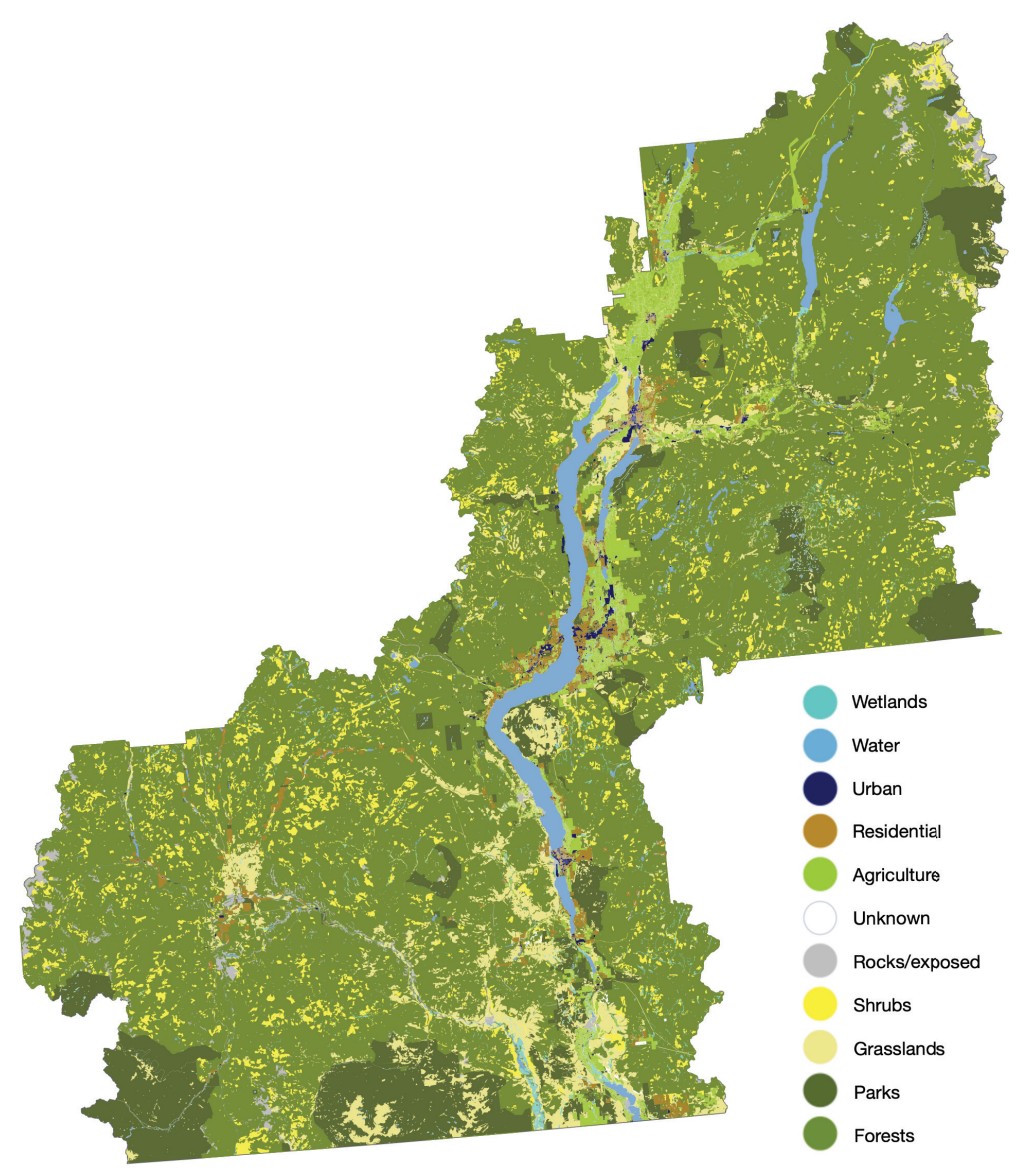

**Appendix 2—figure 1.** Map of the distribution of major LULC types across the case study region. General LULC types in the region include, in decreasing order of area: forests (16,281 km²), grasslands (1482 km²), natural parks (2,403 km²; NB: contains several of the other listed LULC categories), shrubs (1349 km²), agricultural (842 km²), lakes (599 km²), urban residential (220 km²), rural residential (220 km²), wetlands (182 km²), rock/rubble (161 km²), exposed land (113 km²), manicured parks (45 km²), rivers (38 km²), commercial (23 km²), industrial (23 km²), urban institutional (16 km²), and reservoirs (6 km²; *Field et al., 2017*).

# Appendix 3

Summary tables of total areas and proportional coverages for major LULC types within each ES supply area and link type.

**Appendix 3—table 1.** Percent of LULC overlapped by each ES supply area.

| | | | High value supply areas (Top 50%) | | | | | | | |
|---|---|---|---|---|---|---|---|---|---|---|
| | | Study Area | PA | | | WF | | | LA | |
| LULC Type | Total area (ha) | 2158001 | 12,606 | | | 922,425 | | | 1456241 | |
| | | % of study area | Area (ha) | % of supply area | % of total LULC | Area (ha) | % of supply area | % of total LULC | Area (ha) | % of supply area | % of total LULC |
| Forest (incl. parks) | 1,654,215 | 76.65 | 30.85 | 0.24 | 1.86e-03 | 725852.30 | 78.69 | 43.88 | 1354434.00 | 93.01 | 81.88 |
| Park | 255,964 | 11.86 | 4.25e-06 | 0 | 0 | 129588.70 | 14.05 | 50.63 | 190476.70 | 13.08 | 74.42 |
| Grassland | 147,610 | 6.84 | 0.12 | 9.80e-04 | 8.37e-05 | 72050.87 | 7.81 | 48.81 | 3783.53 | 0.26 | 2.56 |
| Shrub | 124,953 | 5.79 | 0 | 0 | 0 | 53425.76 | 5.79 | 42.76 | 3889.10 | 0.27 | 3.11 |
| Agriculture | 79,769 | 3.70 | 12605.44 | 100.00 | 15.80 | 31338.00 | 3.40 | 39.29 | 241.00 | 0.02 | 0.30 |
| Water | 64,272 | 2.98 | 0 | 0 | 0 | 0 | 0 | 0 | 63469.20 | 4.36 | 98.75 |
| Residential | 34,897 | 1.62 | 3.08e-04 | 0 | 0 | 10990.55 | 1.19 | 31.49 | 241.44 | 0.02 | 0.69 |
| Rock / exposed | 27,373 | 1.27 | 2.78e-04 | 0 | 0 | 5387.80 | 0.58 | 19.68 | 779.82 | 0.05 | 2.85 |
| Wetland | 18,207 | 0.84 | 9.44e-06 | 0 | 0 | 18128.12 | 1.97 | 99.57 | 12586.06 | 0.86 | 69.13 |
| Urban | 6187 | 0.29 | 0 | 0 | 0 | 1918.28 | 0.21 | 31.01 | 27.75 | 1.91e-03 | 0.45 |
| Unknown | 1291 | 0.06 | 1.72 | 0.01 | 0.13 | 473.51 | 0.05 | 36.67 | 52.74 | 3.62e-03 | 4.08 |

**Appendix 3—table 2.** Percent of LULC overlapped by each topographic link.

| | | | Topographic links | | | | | | | |
|---|---|---|---|---|---|---|---|---|---|---|
| | | Study Area | WF —> PA | | | WF —> WF | | | WF —> LA | |
| LULC Type | Total area (ha) | 2158001 | 4,079 | | | 44,449 | | | 4,695 | |
| | | % of study area | Area (ha) | % of link area | % of total LULC | Area (ha) | % of link area | % of total LULC | Area (ha) | % of link area | % of total LULC |
| Forest (incl. parks) | 1,654,215 | 76.65 | 2205.94 | 54.08 | 0.13 | 19069.77 | 42.90 | 1.15 | 2688.56 | 57.26 | 0.16 |
| Park | 255,964 | 11.86 | 438.39 | 10.75 | 0.17 | 3146.14 | 7.08 | 1.23 | 682.44 | 14.54 | 0.27 |
| Grassland | 147,610 | 6.84 | 433.05 | 10.62 | 0.29 | 3065.90 | 6.90 | 2.08 | 477.85 | 10.18 | 0.32 |
| Shrub | 124,953 | 5.79 | 102.67 | 2.52 | 0.08 | 1230.22 | 2.77 | 0.98 | 195.80 | 4.17 | 0.16 |
| Agriculture | 79,769 | 3.70 | 807.02 | 19.79 | 1.01 | 2329.46 | 5.24 | 2.92 | 596.03 | 12.70 | 0.75 |
| Water | 64,272 | 2.98 | 127.65 | 3.13 | 0.20 | 15664.49 | 35.24 | 24.37 | 184.83 | 3.94 | 0.29 |
| Residential | 34,897 | 1.62 | 218.51 | 5.36 | 0.63 | 1368.65 | 3.08 | 3.92 | 304.98 | 6.50 | 0.87 |
| Rock / exposed | 27,373 | 1.27 | 46.70 | 1.14 | 0.17 | 671.99 | 1.51 | 2.45 | 112.70 | 2.40 | 0.41 |
| Wetland | 18,207 | 0.84 | 56.51 | 1.39 | 0.31 | 732.48 | 1.65 | 4.02 | 51.79 | 1.10 | 0.28 |
| Urban | 6187 | 0.29 | 43.92 | 1.08 | 0.71 | 185.53 | 0.42 | 3.00 | 45.69 | 0.97 | 0.74 |
| Unknown | 1291 | 0.06 | 1.07 | 0.03 | 0.08 | 2.83 | 0.01 | 0.22 | 1.72 | 0.04 | 0.13 |

**Appendix 3—table 3.** Percent of LULC overlapped by each overlapping links.

| | | Study Area | PA —> LA | | | PA —> WF | | | WF —> PA | | | WF —> LA | | | LA —> WF | | |
|---|---|---|---|---|---|---|---|---|---|---|---|---|---|---|---|---|---|
| LULC Type | Total area (ha) | 2158001 | 8.57024 | | | 4747.30300 | | | 4672.03900 | | | 684,301 | | | 685,377 | | |
| | | % of study area | Area of overlap (ha) | % of link area | % of LULC area | Area (ha) | % of link area | % of total LULC | Area (ha) | % of link area | % of total LULC | Area (ha) | % of link area | % of total LULC | Area (ha) | % of link area | % of total LULC |
| Forest (incl. parks) | 1,654,215 | 76.65 | 2.44 | 28.47 | 1.47e-04 | 31.28 | 0.66 | 1.89e-03 | 29.11 | 0.62 | 1.76e-03 | 621652.40 | 90.84 | 37.58 | 624572.00 | 91.13 | 37.76 |
| Park | 255,964 | 11.86 | 0.37 | 4.31 | 1.44e-04 | 14.13 | 0.30 | 0.01 | 14.04 | 0.30 | 0.01 | 100349.60 | 14.66 | 39.20 | 100349.60 | 14.64 | 39.20 |
| Grassland | 147,610 | 6.84 | 0.24 | 2.75 | 1.59e-04 | 43.72 | 0.92 | 0.03 | 42.58 | 0.91 | 0.03 | 2543.49 | 0.37 | 1.72 | 2543.49 | 0.37 | 1.72 |
| Shrub | 124,953 | 5.79 | 0.02 | 0.18 | 1.27e-05 | 0.64 | 0.01 | 5.15e-04 | 0.64 | 0.01 | 5.15e-04 | 2481.88 | 0.36 | 1.99 | 2481.88 | 0.36 | 1.99 |
| Agriculture | 79,769 | 3.70 | 5.97 | 69.71 | 0.01 | 4652.62 | 98.01 | 5.83 | 4579.59 | 98.02 | 5.74 | 150.98 | 0.02 | 0.19 | 150.98 | 0.02 | 0.19 |
| Water | 64,272 | 2.98 | 0.27 | 3.19 | 4.25e-04 | 0.50 | 0.01 | 7.84e-04 | 0.50 | 0.01 | 7.84e-04 | 2057.04 | 0.30 | 3.20 | 2057.04 | 0.30 | 3.20 |
| Residential | 34,897 | 1.62 | 0.01 | 0.08 | 1.86e-05 | 12.32 | 0.26 | 0.04 | 12.02 | 0.26 | 0.03 | 124.52 | 0.02 | 0.36 | 124.52 | 0.02 | 0.36 |
| Rock / exposed | 27,373 | 1.27 | 0.05 | 0.53 | 1.64e-04 | 5.40 | 0.11 | 0.02 | 5.11 | 0.11 | 0.02 | 291.40 | 0.04 | 1.06 | 291.40 | 0.04 | 1.06 |
| Wetland | 18,207 | 0.84 | 0.16 | 1.92 | 9.02e-04 | 3.83 | 0.08 | 0.02 | 3.82 | 0.08 | 0.02 | 12387.53 | 1.81 | 68.04 | 12387.53 | 1.81 | 68.04 |
| Urban | 6187 | 0.29 | 0 | 0 | 0 | 2.02 | 0.04 | 0.03 | 1.97 | 0.04 | 0.03 | 14.73 | 2.15e-03 | 0.24 | 14.73 | 2.15e-03 | 0.24 |
| Unknown | 1291 | 0.06 | 0 | 0 | 0 | 0 | 0 | 0 | 0 | 0 | 0 | 0 | 0 | 0 | 29.40 | 4.29e-03 | 2.28 |

## Appendix 4

### Discussion of uncertainty

We did not incorporate a measurement of uncertainty into our approach. For example, we did not attempt to directly assess spatial autocorrelation of our functional connectivity models with any other, potentially influential ecological processes, as we were interested in providing straight-forward and replicable rationale for mapping linkages; however, this precluded us from being able to parse the presence of shared drivers and potential artefacts in proxy or primary data. Additionally, the location and value of the identified connectivity corridors may be driven by the assumptions of original ES mapping and the threshold (top 50%) we used to delineate high-value supply areas. Several publications have suggested that incorporating uncertainty measures is necessary for producing reliable results to support decision making and will lead to improved understanding of the system under study through identification of the most compelling findings (*Seppelt et al., 2011*; *Hamel and Bryant, 2017*; *Stritih et al., 2019*). Sources of uncertainty considered in ES assessments are related to models of ecological processes; subjective choices of researchers and/or participants; and practical modelling skills and data quality (*Gos and Lavorel, 2012*; *Crossman et al., 2013*; *Hou et al., 2018*; *Wang et al., 2018*). To date, only a limited number of studies on ES interactions have incorporated measures of uncertainty and/or model validation (*Boerema et al., 2017*; *Agudelo et al., 2020*). As unconfirmed results are difficult to reliably assess, they are not as useful for direct practical applications (*Agudelo et al., 2020*). Studies with the express purpose of providing guidance for on-the-ground multi-ES planning should therefore incorporate metrics of uncertainty and model validation procedures.

# Appendix 5

## Validation of topographic functional connectivity mapping

### Methods

We validated our approach for mapping the functional connectivity between ES supply areas separated in space by executing a null model and comparing these and our topographic link results (all originating in WF supply areas: WF to PA, WF to WF, and WF to LA) with empirical data on surface waterflow. For this, we subset the data to a single subbasin for computational efficiency, and used a random number generator to select a subbasin from those that contained all 3 types of ES supply area polygons, and therefore would have the potential to contain all 3 types of topographic links. We then ran LCP analyses using uniform resistance maps where all cells = 1, so that results approximated straight lines between the supply area centroid and the goal point, the latter of which had the same coordinates as that of the functional connectivity topographic LCP analysis as basin outflow points are well-established. We then executed the same procedures on resulting null model LCP lines that were used to produce topographic corridors between pairs of supply areas (Appendix 7). Following this, we created a buffer equal to grid cell width (29 m) around the perimeters of (a) topographic and (b) null model links. Then, we calculated percent intersection (overlap) of each link in (a) and (b) with BC TRIM water lines data, which served as empirical data for location of surface waterflow. The WF model buffer width was applied to account for potential grid cell alignment and/or spatial resolution mismatches between ours and the TRIM source data (NB: the TRIM dataset is based on smaller cell size as minimum horizontal surface accuracy requirements are 10 m for source DEMs). A 150 m buffer width was applied around TRIM data prior to analyses, which is related to the functional role of riparian ecosystems in water flow regulation (*Fischer and Fischenich, 2000*; *GeoBC, 2003*). To determine whether the topographic corridors included a significant proportion of TRIM waterflow data, we compared the percentage of each topographic link corridor with the percentage of each null model segment within TRIM line data using two-sample Mann-Whitney (or 'Wilcoxon') tests (*Milton, 1964*). This validation method determines whether the functional topographic corridors contained more or less surface waterflow data than expected relative to links produced by a random spatial model, and assumes a relatively high proportion of topographic WF corridor area will co-occur with riparian areas associated with TRIM waterflow data (*Bond et al., 2017*). We did not include link weighting in our validation analyses as comparative empirical data on this is not available for our study area; nor did we incorporate a measure of uncertainty (discussed in Appendix 4).

### Results

Bellevue Creek, a subbasin located in the east-central portion of our study area within the Okanagan watershed, was randomly selected for validation analyses. It is 87.8 km², which covers approximately 0.4% of the total study area. The total number of resulting line segments for functional connectivity vs. null model link creation were 33 and 56 for WF to WF supply areas; 62 and 34 for WF to PA; and 13 and 11 for WF to LA; respectively. For all link types, the data produced by the functional connectivity models co-occurred over significantly more proportional area with TRIM waterflow data relative to links produced with the null model (*Appendix 5—table 1*; *Appendix 5—figure 1*). This confirms that the method for producing topographic links between supply areas presented in our study results in spatially and functionally valid placement of ES connectivity corridors. All validation analyses were completed using R (*R Development Core Team, 2013*) and ArcGIS 10.7.1 (*ESRI, 2011*).

**Appendix 5—table 1.** Summary and validation analyses data for topographic links produced by functional connectivity and null models for Bellevue Creek.

|  | Functional connectivity model | | | Null model | | |
|---|---|---|---|---|---|---|
|  | WF ~ WF | WF ~ PA | WF ~ LA | WF ~ WF | WF ~ PA | WF ~ LA |
| Number of links | 33 | 62 | 13 | 56 | 34 | 11 |
| Total link area (km²) | 9.05 | 18.23 | 5.12 | 23.21 | 20.06 | 3.58 |
| Total overlap area (km²) with BC TRIM data | 5.08 | 12.22 | 3.63 | 3.80 | 3.88 | 0.99 |
| Median % overlap with BC TRIM data | 42 | 67 | 69 | 12 | 25 | 29 |
| Wilcoxon p-value | 1.59e-03 | 5.29e-09 | 2.09e-04 | - | - | - |

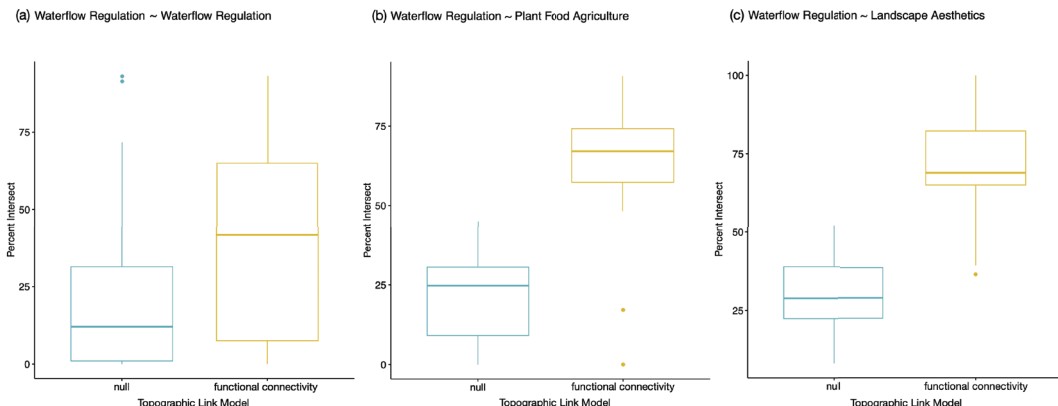

**Appendix 5—figure 1.** Boxplots summarizing percent overlap of buffered BC TRIM data with buffered topographic links between (**a**) WF; (**b**) WF and PA; and (**c**) WF and LA supply areas produced by functional connectivity and null models for Bellevue Creek validation analyses.

# Appendix 6

## Data sources for original ES supply area mapping

**Appendix 6—table 1.** Summary of data sources and rationale for water flow regulation LULC mapping prioritization (from *Field et al., 2017*).

| Map rank | Data source | Source | File name(s) | Year(s) | Type | Fields/Classes | Details |
|---|---|---|---|---|---|---|---|
| 1 | Agricultural Land Use Inventory (ALUI) | BC Ministry of Agriculture (data user agreement) | | 2013 + 2014 | | Cover = ATVC (Anthropogenic Terrestrial Vegetated Cultivated) | Ground-truthed data, not strictly based on remotely sensed image interpretation. Used ATVC category (Agricultural Terrestrial Vegetated Cultivated) class to identify specific agricultural categories – Orchards, Vineyards, Field, Other (2013/14). |
| | | | | | Agriculture - Orchards | CovType = C200 (Tree Fruits) | |
| | | | | | Agriculture - Field | CovType = C320 (Berries) or CovType = C500 (Vegetables) | |
| | | | | | Agriculture - Vineyards | CovGroupType = C311 (Grapes) | |
| | | | | | Agriculture - Other | CovType = C100 (Cereals, Grains, Oilseeds), C400 (Forage, Pasture), C600 (Floriculture), C710 (Specialty), C720 (Turf), C730 (Nut Trees), C810 (Nursery), C820 (Tree Plantations), C900 (Other) | |
| 2 | Okanagan Wetlands Strategy (OWS) | Ecoscape Consulting Ltd. | | 2009 + 2010 + 2011 + 2014 | Wetlands | | Compiled regional database. Includes data from the following sources: City of Kelowna WIM (2009);BC Freshwater Atlas (2014); MOE Wetland Inventory Project (2009); Alkali-Saltgrass Herbaceous Vegetation Community Assessment (2011); SEI/TEM for the study area; SHIM (BX Creek, NORD, Vaseux Creek and Oliver, Prairie Creek, Winfield Creek, various dates); FIM (Kalamalka, Wood, Mabel, Mara, Okanagan, Osoyoos, various dates); LRIM (Lower Shuswap River Inventory and Mapping, 2010); Ducks Unlimited (DU) data (various sources). |
| 3 | BC Freshwater Atlas (FWA) | BC Ministry of Environment (Data Distribution Service) | | 2009 | Lakes | | Lakes |
| | | | | | Rivers | | Rivers (polygons) |
| | | | | | Man Made Waterbodies | | Reservoirs |

*Appendix 6—table 1 Continued on next page*

*Appendix 6—table 1 Continued*

| Map rank | Data source | Source | File name(s) | Year(s) | Type | Fields/Classes | Details |
|---|---|---|---|---|---|---|---|
| 4 | Municipal and Regional Zoning | RDNO Open Data | RDNO | 2017 | Urban - Commercial | | Simplified and compiled zoning data from all municipal jurisdictions in the study area. |
| | | | AreaBZoning | 2017 | Urban - Institutional | Zoning and Zoning_Des | |
| | | | AreaCZoning | 2017 | Urban - Industrial | Zoning and Zoning_Des | Used supporting by-law documentation to limit local codes |
| | | | AreaDZoning | 2017 | Urban - Residential | Zoning and Zoning_Des | to categories: Urban – Commercial, Urban – Industrial, Urban – |
| | | | AreaEZoning | 2017 | Rural - Residential | Zoning and Zoning_Des | Institutional, Urban – Residential, Rural – Residential. (Parks |
| | | | AreaF_OCP | 2017 | Parks | Zoning and Zoning_Des | were also identified in the zoning data but not incorporated until later |
| | | | Coldstream_Zoning | 2017 | | Zoning and Zoning_Des | in the model). |
| | | | Enderby_Zoning | 2016 | | Zoning and Zoning_Des | |
| | | | Lumby_Zoning | 2012 | | Zoning and Zoning_Des | |
| | | City of Vernon Open Data | Vernon_Zoning | 2004 | | Zoning_Val and Zoning_Des | |
| | | RDCO (data user agreement) | RDCO | 2017 | | | |
| | | | BrentTrepanier_FutureLandUse | 2017 | | | |
| | | | Ellison_FutureLandUse | 2017 | | | |
| | | | JoeRich_LandUse | 2017 | | | |
| | | | LakeCountry_FutureLandUse | 2017 | | | |
| | | | Peachland_LandUse | 2017 | | | |
| | | | RuralWestside_FutureLandUse | 2017 | | | |
| | | | SouthSlopes_FutureLandUse | 2017 | | | |
| | | | WestKelowna_LandUse | 2017 | | | |
| | | City of Kelowna Open Data | Kelowna_Zoning | 2017 | | ZoningCode | |
| | | RDOS Open Data | RDOS Zoning | 2017 | | Designation | |

*Appendix 6—table 1 Continued on next page*

*Appendix 6—table 1 Continued*

| Map rank | Data source | Source | File name(s) | Year(s) | Type | Fields/Classes | Details |
|---|---|---|---|---|---|---|---|
| 5 | Sensitive Ecosystems Inventory (SEI) | RDCO Open Data | RDCO_SEI | 2001–2015 | Forest | BW (Broadleaf Woodlands); MF (Mature Forest); OF (Old Forest); WD (Coniferous Woodlands) | RDCO SEI coverage for RDCO portion of study area, provincial SEI coverage for north and south regional districts. Only selected polygons that contained sensitive ecosystems in the primary ground cover field. |
| | | | | | Shrub | SV (Sparsely Vegetated) | |
| | | | | | Grassland | GR (Grasslands) | |
| | | BC Ministry of Environment (Data Distribution Service) | SEI | 2010 | Forest | BW (Broadleaf Woodlands); MF (Mature Forest); OF (Old Forest); WD (Coniferous Woodlands) | |
| | | | | | Shrub | SV (Sparsely Vegetated); AS (Antelope-Brush Steppe); SS (Sagebrush Steppe) | |
| | | | | | Grassland | GR (Grasslands); DG (Disturbed Grasslands) | |
| 6 | Grasslands | Grasslands Conservation Council (GCC) | GCC Grasslands | 2017 | | | Compiled grasslands data. |
| 7 | Vegetative Resource Inventory (VRI) | BC Ministry of Forest, Lands and Natural Resource Operations (Data Distribution Service) | | 2002–2017 | Forest | BCLCS Level 1 (BCLCS_LEVE) = V (Vegetated); BCLCS Level 4 (BCLCS_LEVE3) – TC (Treed Coniferous); TB (Treed Broadleaf); TM (Treed Mixed) | VRI categories: Vegetated – Treed, Vegetated – Shrub, or Vegetated – Grasslands |
| | | | | | Grassland | | |
| | | | | | Shrub | | |
| 8 | Parks | See notes on zoning data | Zoning Data (all jurisdictions) | 2004–2017 | Park_Manicured | See notes on zoning data | Parks were subdivided into two classes – Natural and Manicured. Natural parks included provincial parks and regional parks. Manicured parks included parks from urbanized areas (Penticton, Vernon, Armstrong, RDCO). This included all of RDCO since this zoning mainly covers urban and urban fringe areas. Manicured parks also included any zoned parks in the study area that were coincident with DMTI golf course points data, even if these fell outside of the urban centres. |
| | | Data Distribution Service | Provincial Parks | 2017 | Park_Natural | | |
| | | Data Distribution Service | Conservation Lands | 2013 | Park_Natural | | |
| | | RDCO Open Data | RDCO Parks | 2017 | Park_Natural | | |
| 9 | Vegetative Resource Inventory (VRI) | BC Ministry of Forest, Lands and Natural Resource Operations (Data Distribution Service) | | 2002–2017 | Rock/Rubble | BCLCS Level 4 = Rock/Rubble (1447) | VRI categories: Rock/ Rubble, Snow/Ice, Exposed Land, Water (Lakes, Rivers, Reservoirs) |
| | | | | | Rivers | BCLCS Level 4 = Water (1069) | |
| | | | | | Lake | BCLCS Level 4 = Water (1069) | |
| | | | | | Exposed Land | BCLCS Level 4 = Exposed Land (865) | |
| | | | | | Reservoir | BCLCS Level 4 = Water (1069) | |

**Appendix 6—table 2.** Calculated and assumed NDVI results, infiltration rates (0%–100%), and qualitative aesthetic valuations for each LULC type.

Blank (grey) NDVI values were not calculated; therefore, and infiltration rate was assumed (from *Field et al., 2017*).

| LULC | Area (ha) | NDVI values | | | | | Infiltration % | Aesthetic Value |
| | | Min | Max | Range | Mean | Stdv | | |
| --- | --- | --- | --- | --- | --- | --- | --- | --- |
| Roads | | | | | 0.1000* | | 0 | |
| Urban - Industrial | 2,302 | −0.2329 | 0.6099 | 0.8428 | 0.1614 | 0.1040 | 19 | No Data |
| Urban - Commercial | 2,319 | −0.3094 | 0.6194 | 0.9287 | 0.2000 | 0.1369 | 32 | No Data |
| Rock/Rubble | 16,102 | −0.3993 | 0.6201 | 1.0194 | 0.2149 | 0.1163 | 36 | 319 |
| Urban - Residential | 21,952 | −0.1594 | 0.6142 | 0.7736 | 0.2410 | 0.0868 | 45 | 318 |
| Urban - Institutional | 1,566 | −0.2074 | 0.6349 | 0.8423 | 0.2524 | 0.1309 | 48 | No Data |
| Exposed Land | 11,272 | −0.2332 | 0.6792 | 0.9124 | 0.2528 | 0.1197 | 48 | No Data |
| Rural - Residential | 21,952 | −0.1531 | 0.6326 | 0.7857 | 0.3025 | 0.0851 | 64 | 318 |
| Agricultural | 35,291 | −0.1763 | 0.6573 | 0.8335 | 0.3151 | 0.1010 | 68 | 268 |
| Agriculture - Field | 51 | 0.0494 | 0.6297 | 0.5803 | 0.3187 | 0.1017 | 69 | |
| Unknown | 1,291 | 0.0000 | 0.6084 | 0.6084 | 0.3264 | 0.0908 | 72 | No Data |
| Agriculture - Vineyards | 4,335 | 0.0061 | 0.6189 | 0.6128 | 0.3311 | 0.0742 | 73 | 275 |
| Agriculture - Other | 33,930 | −0.0636 | 0.6704 | 0.7340 | 0.3637 | 0.1231 | 83 | |
| Agriculture - Orchards | 5,802 | −0.0364 | 0.6349 | 0.6713 | 0.4164 | 0.0943 | 100 | |
| Park - Manicured | 4,534 | | | | | | 100 | 205 |
| Forests | 1628131 | | | | | | 100 | 664 |
| Grasslands | 148,236 | | | | | | 100 | 319 |
| Shrub | 134,873 | | | | | | 100 | 319 |
| Wetland | 18,207 | | | | | | 100 | 661 |
| Lake | 59,911 | | | | | | 100 | 1000 |
| River | 3,763 | | | | | | 100 | 1000 |
| Reservoir | 598 | | | | | | 100 | 1000 |

* = assumed NDVI value for paved roads based on impervious threshold (https://phenology.cr.usgs.gov/ndvi_foundation.php).

No Data = no value applied in mapping due to lack of data.

Gray cells = not included in analyses.

## Appendix 7

### Methodological details

### Rationale and procedure for delineating top-value ES supply areas

Studies that consider multiple ES have found that the distribution of at least one ES may be ubiquitous across a regional landscape (e.g. *Queiroz et al., 2015*), and/or isolated supply areas may be present within a non-ES-provisioning matrix (e.g. *Qiu and Turner, 2013*; *Figure 2*). Both situations were true for our landscape based on the ES selected, so for the purposes of creating supply areas, we chose to only retain areas with supply values above a top 50% threshold. This was because two of the ES types we selected, WF and LA, had near-ubiquitous spatial coverage with values ranging from very low to very high, and because management applications often are most interested in maintaining the highest-value provisioning areas (e.g. *Turner et al., 2007*). For PA, we subset the top 50%-valued polygons from the original mapping; for WF and LA, we subset the top 50%-valued raster cells of each of the regional ES maps, then converted these cells to single-part polygons based on aggregating adjacent cells within a diagonal raster cell width (~29 m). We did not assess the potential sensitivity of ES supply area distribution or connectivity results to the threshold choice as the primary goal of this study was to provide a methodological proof of concept, rather than to precisely map ES supply or connectivity, though future research concerned with the latter should test sensitivity of threshold value choices.

   Due to the large file size of the waterflow regulation (WF) data, supply area delineation steps were run separately for identified sub-basins (n = 118) in our case study area. To identify sub-basin catchment areas, we used BC Major Watershed, Fresh Water Atlas (FWA) Watersheds, and FWA Streams datasets (*FLNRO et al., 2017*; see *Field et al., 2017* for data source descriptions; datasets available at https://www.data.gov.bc.ca/). For major watersheds with significant (or complete) overlap with our study area, which would result in a large number of within-basin ES supply areas and therefore potentially lead to computational limitations, nested sub-basins were identified. These major watersheds included Kettle (west), Okanagan, Similkameen, and South Thompson rivers (*Appendix 7—figure 1*). FWA Watersheds with a common terminus into valley bottom waterbodies, verified using the FWA Streams dataset (*FLNRO et al., 2017*), were merged. Several of the major watersheds (Columbia, Fraser, Kettle (east), Thompson, and Washington (Coast) rivers) overlapped with our study area primarily along its border; the overlapping portions of these watersheds were clipped and added to the sub-basin dataset (see *Appendix 7—figure 1B*). The high-value WF raster was then split by sub-basins using the nearest neighbour sampling technique and Split Raster tool in ArcMap. Each major watershed was assigned a unique 'goal' point location for LCP analyses, that is, sub-basins within a major watershed shared the same goal point. In some cases, the mapped borders of the BC Major Watersheds and FWA Watersheds did not exhibit perfect overlap; therefore, following the high-value WF split exercise, the ArcGIS Erase tool was run by erasing the spatial extent of sub-basins from other BC Major Watersheds where they overlap. This ensured that WF supply areas were assigned to sub-basins by prioritizing the more detailed FWA Watershed dataset.

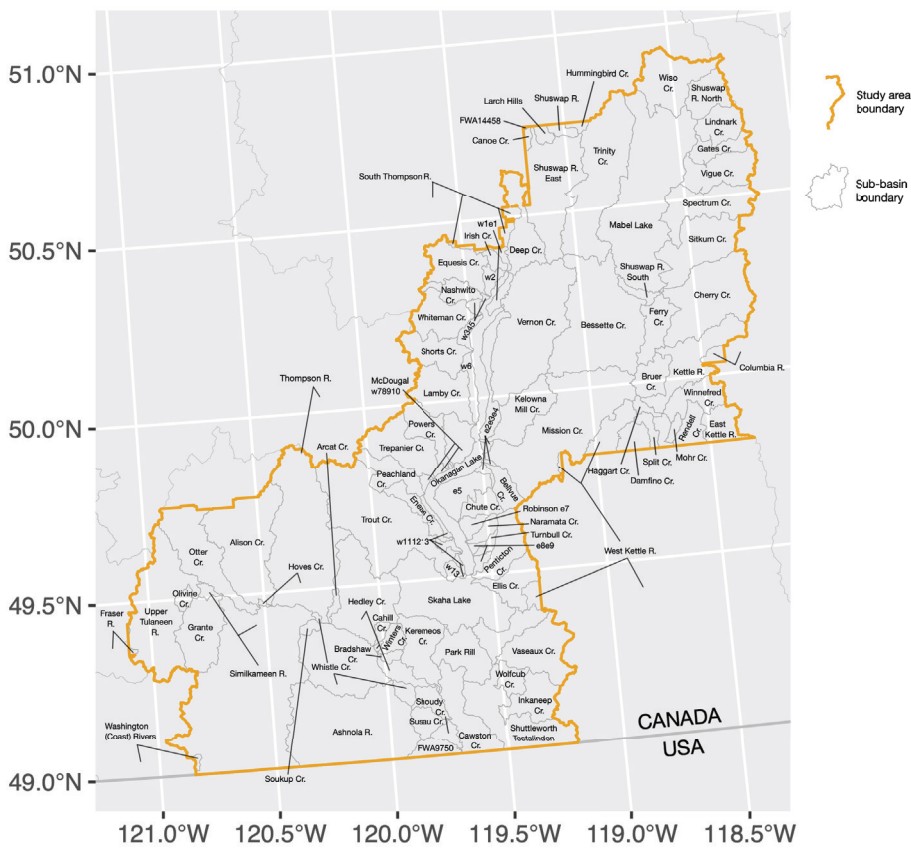

**Appendix 7—figure 1.** Map of major watersheds and sub-basins within and surrounding the case study landscape in southern interior British Columbia, Canada.

## Detailed methods for building pairwise overlapping and topographic links

Theory on the mechanisms between *overlapping* pairs of ES supply areas was as follows. For PA, the presence of vegetation crops can contribute to WF through providing a variety of beneficial ecological properties (e.g. soil texture, low-slope, high-perviousness, floodplains, riparian areas, and seasonally flooded fields; *Power, 2010*), although the weight of this positive interaction may be higher if agricultural land was allowed to return to a natural vegetated state (*Roa-García et al., 2011*). PA also interacts with LA by providing farmland that is recognized as being aesthetically valuable (e.g. vineyards; *Wagner and White, 2009*; *Field et al., 2017*) where these areas overlap. For WF, a direct positive influence stems from the spatial confluence of high-value WF areas on PA and LA supply areas through the maintenance of underlying hydrological processes where they co-occur (e.g. *DeLaney, 1995*; *Nelson et al., 2009*; *Seavy et al., 2009*). In the other direction, high-value terrestrial LA supply areas can be linked to WF areas through supportive ecological functions (e.g. pervious and water-retaining vegetated landscapes, floodplains in populated areas; *Boyd and Banzhaf, 2007*; *Van der Ploeg et al., 2010*; *van Berkel and Verburg, 2014*; *Carpenter et al., 2015*; *Table 1*).

To build *topographic* links between pairs of ES supply areas, first a separate least cost path (LCP) analysis was run for each WF supply area polygon to identify link corridors between these and other ES supply areas (*Figure 6a*). LCP analysis is a common method of mapping directional ecological corridors in landscape connectivity research, where the movement of an organism (or abiotic unit) is simulated across a resistance (cost) surface from a start to a destination point, and the lowest-

accumulated resistance becomes the most likely path it will follow across a landscape (*Beier et al., 2009*). LCP can also be effectively used for hydrological flow models, where the algorithm seeks to minimize cumulative elevation along its path (e.g. *Melles et al., 2011*). The starting WF supply area polygon centroid was used as the 'origin' point for each associated LCP analysis. A single LCP 'goal' point was determined for each sub-basin by identifying the basin stream outlet (*FLNRO et al., 2017*); the LCP goal point coordinates were identified as the intersect of this stream line feature and the valley-bottom line feature of the associated major watershed (*Field, 2021*). If multiple outlets were present in a sub-basin (e.g. Okanagan sub-basin 'w11213'), the furthest downstream outlet line feature was used. LCP transition functions were built based on the assumption of downslope waterflow over a DEM surface, and allowed for connecting to a 16-cell neighborhood to avoid paths being terminated based only on a single depression cell (*van Etten, 2017*).

Once initial LCPs were created, segments of LCP lines were erased where they were overlapped by a non-origin ES supply area polygon, and resulting disconnected lines were made into separate line features. Next, we deleted any lines that were deemed invalid from the perspective of real-world ES connectivity. Specifically, any resulting lines that intersected with the sub-basin goal point were deleted as we were only interested in retaining connections between pairs of ES supply areas. Certain ES polygons with an irregular shape had a centroid external to their polygon coverage, which resulted in line segments that initiated at the origin polygon centroid and terminated on the origin polygon border; these were also deleted as they did not represent links between a pair of ES supply areas. Irregular shaped nodes also sometimes yielded lines that were connected between two points on the parent-polygon border. These were retained to account for functional maintenance feedback connections within a supply area; however, we ensured that any such lines associated with two borders of an intersected polygon (i.e. where the line segment was part of a non-origin polygon) were deleted to avoid duplication with feedback links identified when separate analyses were run with the intersected polygon (in this example) as the origin (*Appendix 7—figure 2b*).

We then identified influential landscape features (ILFs) as additional WF polygons downstream of each sub-basin in the valley-bottom and associated with wetlands, floodplains, riparian areas, and/or seasonally flooded fields, which are functionally linked to upstream hydrological regulation. We connected ILFs to each upstream sub-basin outlet point, and individually merged these sub-basin lines with the LCPs for each WF supply area within that sub-basin (*Appendix 7—figure 2*). Additionally, if a sub-basin flowed into a lake or reservoir, all ILF polygons immediately adjacent to that waterbody were included in the list of 'downstream' supply areas. Lastly, because the DEM raster resolution was approximately 29 m, some LCPs flowed outside sub-basin boundaries between the origin polygon (typically those close to a sub-basin boundary) and the goal point. Any nodes outside the sub-basin of interest that were overlapped by such LCPs were not considered to be true 'intersections' and such line segments were therefore excluded from within sub-basin links. However, such LCPs were still able to become connected to downstream ILFs.

For topographic corridor mapping we needed to address various rare analytical outcomes that became evident upon manual model validation. When we manually inspected preliminary spatial results, we found that the LCP analyses resulted in some connections that violated landscape topography. For example, some LCPs from WF supply areas associated with relatively flat lands in the headwaters of one sub-basin (Bellevue Creek) were found to flow south to Okanagan Lake rather than flowing north as they would in reality (*Appendix 7—figure 1*). This was due to the necessity of balancing the smallest possible raster resolution (20 m) with the overall large size of the study area for computational efficiency. As individual sub-basins were analyzed separately for topographic links, invalid linkages between sub-basins were not possible; however, we manually inspected all LCP results and removed any LCPs that violated downslope flow logic from subsequent analyses. Further, we did not incorporate certain rare spatial occurrences. These included instances of a smaller supply area inside bigger one; centroids captured by the incorrect buffer due to two or more centroids occurring close to one another; and LCP segments that resulted in feedback loops that occurred across non-origin supply areas (these loops were likely retained as feedback loops when such nodes served as LCP origins).

We quantified the weight of connections between ES supply areas based on assumptions around the functional relationships between ES, which we discuss below. Since some of the ecological characteristics of both PA and AE areas can support the maintenance of WF supply, and high-

value WF lands maintain the hydrological characteristics that help support PA provision and AE, we assumed that the value of these connections would approximate the available WF capacity. Therefore, the overlapping links between WF and PA, and between WF and LA, were weighted by the summed WF raster values therein. The plant food-providing agricultural areas of the Okanagan are prized by some for their beauty (*Wagner and White, 2009*; though this subjective evaluation is complicated, see *Wagner, 2008*), and we assumed the level of this significance to be equivalent to the underlying LA model value; therefore, such overlap links were unidirectionally weighted from PA to LA by the summed LA raster values therein. We assumed the contribution of upslope, high-value WF lands to the hydrologic maintenance of intersected downslope PA, WF and LA areas to be equivalent to the amount of flow regulation provided by landscape where water flows between these areas. Therefore, we quantified the weight of these unidirectional links by the cumulative value of all WF raster cells on the original map (i.e. not just top ES model values; *Figure 3c*) traversed along corridors. All link weights were obtained by extracting summed raster cell values coincident with overlap link areas or with topographic link segments, then normalized on a unit-less scale from 1 to 10,000. Additionally, raster overlay analysis was conducted, wherein cell values were summed across all eight link types to produce a weighted distribution map of all multi-ES connections for our entire study region.

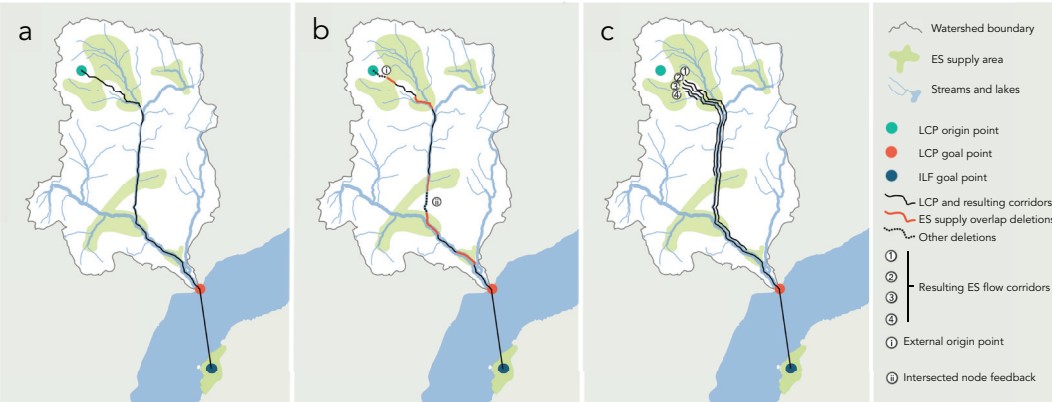

**Appendix 7—figure 2.** Schematic outlining steps for the creation of topographic ES corridors from each origin ES supply area to downslope supply areas. (**a**) An initial line feature resulting from a least cost path (LCP) analysis, that is, from the origin to the goal point, amalgamated with a line from the goal point to a downstream influential landscape feature (ILF). (**b**) Types of LCP segment deletions addressed, including (red) segments overlapped by ES supply area polygons, (**i**) segments from origin points external to origin ES polygon, and (**ii**) segments flowing between two areas of an intersected (i.e. non-origin) ES supply area. (**c**) Resulting ES flow corridors after deletions, including feedbacks to origin ES supply area (4), flows to downslope ES supply areas (1) and (3), and flows to downstream ILF areas (2).

## Appendix 8

### R script files (available on the OSF; *Field, 2021*)

1. TOP-VALUE SUPPLY AREA NODE CREATION (file name: 'nodes_waterflow_FORMAT.R')
2. OVERLAP LINK CREATION (file name: 'links(overlap)_waterflow <> plantag_MASTER_FORMAT.R')
3. SUPPLY AREA CENTROID CREATION (PREP FOR LEAST COST PATH 'LCP' ANALYSES) (file name: 'links_waterflow_ALL_centroids_FORMAT.R')
4. CREATE LCP GOAL POINTS FOR EACH SUB-BASIN/MAJOR WATERSHED (file name: 'links_waterflow_ALL_LCP goal pts_FORMAT.R')
5. LEAST COST PATH 'LCP' ANALYSIS FOR TOPOGRAPHIC LINK CREATION (file name: 'links_waterflow_okanagan_LCPs_SUBBASIN_FORMAT.R')
6. TOPOGRAPHIC LINK CREATION (file name: 'links_waterflow_okanagan_1_MASTER_FORMAT.R')
7. INFLUENTIAL LANDSCAPE FEATURES (ILF) LINKS (file name: 'links_waterflow_okanagan_LCPs_ILF_FORMAT.R')

