## [Editor Report]

Ecosystem services such as agriculture and waterflow regulation may interact, but the nature of these interactions is not well understood.This manuscript proposes a new framework based on approaches from geographic information science (GIS) to assess functional connectivity of ecosystem services, which reveals unexpected links across services and spaces. This paper is of interest to researchers in the fields of ecosystem services and landscape ecology, and more broadly to scientists studying sustainable practices affecting ecosystems.

---

## [Decision Letter]

**Decision letter after peer review:**

Thank you for submitting your article "Mapping the functional connectivity of ecosystem services supply across a regional landscape" for consideration by *eLife*. Your article has been reviewed by 2 peer reviewers and a Reviewing Editor, and the evaluation has been overseen by Meredith Schuman as the Senior Editor. The following individuals involved in review of your submission have agreed to reveal their identity: Cesar Augusto Ruiz Agudelo (Reviewer #1); Maria Dossantos (Reviewer #2).

The reviewers have discussed their reviews with one another, and the Senior Editor has drafted this to help you prepare a revised submission.

Essential revisions:

1) Most of the essential revisions address concerns regarding explanation and validation of the analysis approach raised by all three reviewers. Reviewer 1 points out that the number of ecosystem services studied is small and that it is important to delineate the limitations of the approach. Both other reviewers have questions about choices made for analysis which require clarification both in order to understand limitations, and also for the sake of reproducibility as well as for the potential uptake and application of such approaches by others.

1a. Some additional analyses must be conducted in order to assess the validity of the approach and its results for the chosen dataset.

Reviewer 2: There is very little description on how (i) ecosystem attributes were mapped to ecosystem service supply, (ii) which data sources were used, why and what for and whether there are interlinkages between datasets that make the interpretation of the results less clear, and (iii) there is no description of the ES models – all in appendix but should be in the main text as this is the foundation of the functional connectivity analyses.

Reviewer 2: Functional connectivity is perhaps one of the most difficult aspects to validate, and landscape ecological literature has shown this for quite some time. Indeed, the authors mention the aspect of validation, but maybe it would be possible to compare the results with a null or a random walk model to see at least how comparable the results are?

1b. The authors should clarify the purpose of their analysis. Is this to discover mechanisms, or to evaluate the evidence for mechanisms which are hypothesized a priori?

Reviewer 3: Although the authors aimed to provide answers to "what and where are the mechanisms responsible for maintaining these connections and, consequently, supporting the production of multiple ES" (lines 143-146). I did not get clear information on these "mechanisms" after the reading. These "mechanisms" seems were only first defined by the authors (the two types of links of ES) and then put into the calculation and mapping.

1c. The authors have also chosen an analysis method which may be especially tailored to water flow (WF) and less suitable for other variables, especially when considering their evaluation of topographic (versus overlapping) links.

Reviewer 1: While an interesting proof-of-concept, it is clear that some ecosystem services are more likely than others to exhibit functional connectivity as described in this paper. For instance, the chosen waterflow regulation is necessarily a "flow" (thus exhibiting functional connectivity) while landscape aesthetics might not be.

Reviewer 2: While yes LCP has been widely used in functional connectivity and hydrological studies, does this make it more likely to obtain certain types of results for waterflow regulation than for other ecosystem services? I would recommend the authors to discuss whether the choice of method for functional connectivity matters, and if there are certain methods that work best for one or another ecosystem services. The authors do acknowledge in the discussion that there are other methods, however, also justify the choice based on waterflow thus maybe making it more unlikely to find functional connectivity for other ecosystem services that may be more "corridor" or "resistance" based and thus would be better modeled with other methods?

Reviewer 3: For example, different lands of plant agriculture (PA) can be connected by pollinators, which forms the connection PA-PA. This further points to problems with the calculation of topographic links. The weight of topographic link associated with WF was calculated as the summed WF values of the cells connecting two locations. That means they used related ES values to calculate corresponding ES link. This approach may miss important processes that truly connect ESs at different locations but poorly represented by the related ES values (e.g., pollinator movement are affected by land cover of forest and grassland) and would be difficult to derive useful mechanisms underlying the links.

1d. It is not clear why the analysis of overlapping links is asymmetrical, or why this should be the case.

Reviewer 3: Overlapping links were defined as "areas where the supplies of two different types of ES occur the in the same location on the landscape" (lines 371-372). Based on this definition, overlapping links should be not directional, i.e., the same for both ES link A to B and ES link B to A. But the presented calculations (Table 1) are directional, i.e., overlapping links differ between ES A to B and ES B to A.

1e. It is not clear why the authors use different rules to calculate the overlapping weights for different ES pairs (lines 477-488), nor how such rules are derived (i.e., how should someone trying to reproduce this analysis derive the rules).

Reviewer 3: I do not understand why different rules should be applied to different ES pairs, and how would that affect the results. I wonder if those complicated steps were necessary, if we developed the calculation consistent with the definition (i.e., non-directional).

2) What guidelines can the authors provide to generalize their proof-of-principle approach for future adopters and for others attempting similar analyses? To what extent is the chosen approach applicable to other ecosystem services than those three which the authors have chosen? Might other methods such as those brought up in 1), and mentioned by the authors, be more appropriate for other constellations of ecosystem services? Can the authors suggest how future analyses could incorporate ES dynamics?

Reviewer 1: It is important to delve into the limitations of its approach, its potential replicability, the information needs, and how this limits its use. Would your methodological approach have problems with the incorporation of more ecosystem services? Static scenario (snapshot). In future research, could your methodological proposal overcome this difficulty? If not, how could it be overcome (please generate recommendations based on evidence and literature)?

Reviewer 2: In Table 2 the authors propose the potential functional links (and the lack thereof) for their three ecosystem services. How robust is therefore the framework for ecosystem services for which we do not know whether or not there are functional connectivity links and how can we validate this approach?

3) Clarity and organization of text. Essential revisions under (1) and (2) already address this; however, I (the senior editor) note that the manuscript is already very long and although *eLife* generally does not impose strict length limitations, I do think this is an indication of some need for restructuring. Specifically, the introduction and discussion are disproportionately long, while essential details are missing from methods and results, as outlined above; and at the same time, some details which may be interesting only for specialists take up substantial space in the methods.

3a. Please streamline the introduction and the discussion to allow readers some mental space for the missing essential information. For the introduction, please consider what information readers need in order to understand the importance of your work and to evaluate your work specifically.

3b. Appendices are a great tool to streamline the main text, but please consider that they should not be used not for information which is essential for most readers to understand your analysis and results. They should rather be used for important details that will be needed by specialists.

Reviewer 2: I think in general the writing is very well done, and my main suggestion would be to reduce some unnecessary detail in the methods and in the discussion. More specifically:

– Lines 251-259: do you need to detail the area covered by each land use type in such detail?

– Some descriptions of the methodologies are too detailed, and could be better elaborated in an appendix with the technical aspects. Keep in the main text the descriptions of the steps necessary for the LCP analyses and the identification of functional connectivity. I would suggest to move technical aspects on how the data was prepared for the LCP and of the deletion of segments is (in my opinion) too detailed.

– Move limitations assessment to the main body of the text and explain how did you address/minimize/cope with these limitations within your analyses and results.

3c. Regarding the discussion, I believe that addressing some of the concerns brought up in (1) and (2) will allow it to be more targeted. See also detailed comments from Reviewer 2.

Reviewer 2: It would be important that the authors stay close to the what the results mean and their interpretation, that they discuss the implications of their choices, and would also recommend reducing some of the text of the discussion for a better focus. While there are four ways this study could be expanded to: (i) test whether LCP results differ from random or null expectations for the three ecosystem services, (ii) expand on the number of ecosystem services, (iii) test different functional connectivity metrics and models, and (iv) validate functional connectivity, I am aware that would be several additional studies. I would therefore suggest focusing a few analyses on the first point, to test whether the results presented herein are sensitive to the choices for ecosystem services and differ from random or null expectations [note from senior editor: see Essential Revision 1]. The other expansions could be, at the moment, discussed – for which I would recommend reducing and replacing some discussion content, which I will detail below.

– Lines 691-693: this seems to result from the fact that the method minimizes big elevation differences correct? Can you explain when and when not this could be driven by the method alone?

– Line 698: why riparian areas?

– Lines 711-727: maybe explain these results so that they don't repeat figure 8 – what is the most common LULC within all the pairs, does it differ between the topographic and the overlapping pairs? Does it differ between the pairs of ES? How much of these contributions are a function of LULC entering the models for ES and the weights attributed to LA, PA and WF?

– Lines 879-914 – to me these are off topic and would greatly reduce to the main aspects that allow interpret your findings and implications. These can and should be replaced by a section on the limitations.

– Lines 921-948 and lines 950-967: again can be reduced – I understand you want to explain the implications of your findings but I would first focus on interpreting them, provide the limitations and what is that we learn from it, and then make some inferences of what implications might be – there seems to be several types of implications and maybe you can focus on (i) method and data implications – you already mention that this is a first analysis of ES functional connectivity which is quite new – but does it work for which data and for which method, and for which ES model?, (ii) ES research implications – this is very important – spatial relationships have been used to understand synergies and trade-offs and here you bring a new light to it – where could this lead to new knowledge about ES? and (iii) landscape and conservation planning implications – these are obvious if we want to maintain ES flows and conserve and plan development that does not break this flow. There are bits of each scattered throughout the discussion and would be helpful to consolidate these.

– Line 987-989: why is this a disconnect?

– Lines 1057-1059: I completely agree and would encourage that the authors discuss their results with this in mind.

– Line 1071: it came to my attention at this point that the reference provided would not be the best reference for this point – so I would encourage the authors to check this.

– Lines 1082-1104: can this be consolidated? Is there a need to explain how other methods work or would it be more informative to know what else would the other methods provide that is different from what is provided within the context of your proposed approach?

– Line 1126: why focus on the visualization but not on the outcomes of what functional connectivity is? In general, I would focus the Discussion section more.

*Reviewer #1:*

Modeling the interactions among multiple ecosystem services (ES), should improve our understanding of the co-benefits that ecosystems can provide for human well-being. This contribution presents an approach to address the research gaps of interactions among multiple ecosystem services, building on existing ES mapping and modeling and rooted in landscape connectivity theory, to demonstrate how functional relationships between multiple ES can be represented in the context of connectivity planning across a regional heterogeneous landscape.

This work is an important step in formally addressing the interactions among multiple ecosystem services. The applied landscape connectivity approach is an intelligent and viable route to understand these relationships in complex contexts. The analysis of the information sources is precise and allows the development of a robust methodological proposal to understand these relationships. However, this research continues with partially address these relationships. Insisting on the peer evaluation of ecosystem services is a weakness to be reviewed in more detail. Another point to improve in future research is related to the modeling tools for these interactions. It is essential to overcome static (snapshot) approaches. However, this contribution is an important step to advance in this direction.

Future research needs to involve dynamic environmental modeling approaches (Semantic Metamodeling or the Bayesian Network approach). It is necessary to take a step towards this dynamics understanding. This contribution is a necessary first step.

I have found your work interesting, original, and very useful in advancing the understanding of interactions among multiple ecosystem services.

1. I find the manuscript to be well-written and accurate.

2. The management of the information sources is impeccable.

3. The methodological processes applied are novel and replicable in multiple contexts.

4. The results are consistent and represent a significant advance in understanding interactions among multiple ecosystem services.

I am aware that the analysis of these interactions is an open, complex, and under construction topic. For the above reasons, the contribution is relevant to the field. However, some aspects should be deepened in the manuscript discussion:

1. The number of ecosystem services involved is still smaller (three). How to advance the inclusion of more ecosystem services in future research? Would your methodological approach have problems with the incorporation of more ecosystem services?

2. Static scenario (snapshot). In future research, could your methodological proposal overcome this difficulty? If not, how could it be overcome (please generate recommendations based on evidence and literature)?

3. It is important to delve into the limitations of its approach, its potential replicability, the information needs, and how this limits its use.

Strengthening these aspects in your discussion would be wonderful to strengthen your contribution.

*Reviewer #2:*

Summary:

Field and Parrott set to measure connectivity for a selected set of ecosystem services as a proof of concept for an approach to assess functional connectivity. With a well elaborated analysis of connectivity using least cost path analysis, the authors are able to show landscape multifunctional linkages for three ecosystem services (plant agriculture, waterflow regulation and landscape aesthetics), otherwise missed with more traditional mapping methods. The authors find "hidden" connections between waterflow regulation and the other two ecosystem services, which demonstrate the more general importance of this one ecosystem services for the provisioning of the others. On the other hand, for the other two services there are not so strong connections with the other ecosystem services, which might be due to a different nature of flow or simply because such connections may not exist. With this approach, the authors argue that it becomes possible to understand landscape processes of ecosystem services.

The proposed approach described in this paper has several strengths:

– It addresses a topic that has yet to be further developed within the field of ecosystem services. While there are theoretical concepts regarding how ecosystem services are linked and that they might exhibit landscape and spatial connections, the authors here provide an approach and empirical application of such concept.

– It is an interesting proof-of-concept, and the authors clearly find spatial connectivity within and between ecosystem services, in particular between waterflow and the other two services.

– It combines a set of theoretical underpinnings of ecosystem services and landscape ecology that have yet to be further explored thus advancing the two fields in novel directions.

– It connects both the supply and flow aspects of ecosystem services, and shows how ecosystem services flows are different from ecosystem services connectivity.

The conclusions of this paper are mostly well supported by data, but some aspects of the foundation of the analyses need to be clarified and extended.

– While an interesting proof-of-concept, it is clear that some ecosystem services are more likely than others to exhibit functional connectivity as described in this paper. For instance, the chosen waterflow regulation is necessarily a "flow" (thus exhibiting functional connectivity) while landscape aesthetics might not be. It would be very helpful that the authors discuss whether there are (or not) ecosystem services that are "flow" (i.e. in the functional connectivity aspect, that depend on it value upstream and that dictate the value of the ecosystem service downstream), and others that are not and what are the implications of such to their analytical framework and the findings. Because of this, are there some ecosystem services that could not be analyzed with this approach and why? How does it differ if these ecosystem services are intrinsic, instrumental or relational (following IPBES framework-which I would recommend the authors to use instead of the Millenium ecosystem assessment one)? More specifically, I would like to see a stronger argument for why these three ecosystem services (and not others) that co-occur in the study area and that the authors are quite likely knowledgeable about.

– Choice of connectivity model – the authors chose least path cost analysis. While yes LCP has been widely used in functional connectivity and hydrological studies, does this make it more likely to obtain certain types of results for waterflow regulation than for other ecosystem services? I would recommend the authors to discuss whether the choice of method for functional connectivity matters, and if there are certain methods that work best for one or another ecosystem services. The authors do acknowledge in the discussion that there are other methods, however, also justify the choice based on waterflow thus maybe making it more unlikely to find functional connectivity for other ecosystem services that may be more "corridor" or "resistance" based and thus would be better modeled with other methods? Probably the authors have a good rational and understanding whether or not this is a good choice for other ecosystem services and would be great to include this in the analysis as we would gain a much better understanding of the limitations and opportunities provided by the framework across different types of ecosystem services. These would be very interesting aspects for the *eLife* readership that would want to mimic or think about whether the approach proposed herein could be applied to their system or set of ecosystem services.

– Data analyses: there is very little description on how (i) ecosystem attributes were mapped to ecosystem service supply, (ii) which data sources were used, why and what for and whether there are interlinkages between datasets that make the interpretation of the results less clear, and (iii) there is no description of the ES models – all in appendix but should be in the main text as this is the foundation of the functional connectivity analyses.

– Validation – functional connectivity is perhaps one of the most difficult aspects to validate, and landscape ecological literature has shown this for quite some time. Indeed, the authors mention the aspect of validation, but maybe it would be possible to compare the results with a null or a random walk model to see at least how comparable the results are? Further, in Table 2 the authors propose the potential functional links (and the lack thereof) for their three ecosystem services. How robust is therefore the framework for ecosystem services for which we do not know whether or not there are functional connectivity links and how can we validate this approach?

– Limitations: why are these in an appendix and not in the main body of the text? I would suggest moving these to the main text as this enables more transparency about the analyses.

I do think that the authors have set themselves to a very challenging task, and they do provide a proof-of-principle analysis of why is functional connectivity and how to measure it for a selected set of ecosystem services. However, it would be important that the authors stay close to the what the results mean and their interpretation, that they discuss the implications of their choices, and would also recommend reducing some of the text of the discussion for a better focus. I think this is all very possible, and with the approach proposed herein it becomes therefore possible to understand landscape processes of ecosystem services. These are new advances for the field with the capacity to provide important information on the spatial configuration of ecosystem service flow, which can be very useful for both fundamental and applied understanding of ecosystem services and their relations.

Improved or additional experiments:

– While there are four ways this study could be expanded to: (i) test whether LCP results differ from random or null expectations for the three ecosystem services, (ii) expand on the number of ecosystem services, (iii) test different functional connectivity metrics and models, and (iv) validate functional connectivity, I am aware that would be several additional studies. I would therefore suggest focusing a few analyses on the first point, to test whether the results presented herein are sensitive to the choices for ecosystem services and differ from random or null expectations.

– The other expansions could be, at the moment, discussed – for which I would recommend reducing and replacing some discussion content, which I will detail below.

– Further in Line 370: why this threshold and what are the implications of this choice on the results? Would be nice to have a sensitivity analysis on this threshold.

Recommendations for improving the writing and presentation:

I think in general the writing is very well done, and my main suggestion would be to reduce some unnecessary detail in the methods and in the discussion. More specifically:

– Lines 251-259: do you need to detail the area covered by each land use type in such detail?

– Some descriptions of the methodologies are too detailed, and could be better elaborated in an appendix with the technical aspects. Keep in the main text the descriptions of the steps necessary for the LCP analyses and the identification of functional connectivity. I would suggest to move technical aspects on how the data was prepared for the LCP and of the deletion of segments is (in my opinion) too detailed.

– Move limitations assessment to the main body of the text and explain how did you address/minimize/cope with these limitations within your analyses and results.

*Reviewer #3:*

This study describes approaches from the field of landscape connectivity to map spatial and functional connectivity of multiple ecosystem services (ES). The ideas are interesting, and knowing about the spatial connectivity or links of ES is important both for ecological research and management.

There are some concerns about the approaches developed to define and calculate the links. The results are quite descriptive. It is difficult for me to get any simple conclusion. Although the authors aimed to provide answers to "what and where are the mechanisms responsible for maintaining these connections and, consequently, supporting the production of multiple ES" (lines 143-146). I did not get clear information on these "mechanisms" after the reading. It appears that these "mechanisms" were only first defined by the authors (the two types of links of ES) and then put into the calculation and mapping.

The authors defined two types of links, overlapping and topographic links. (1) For overlapping links, the definition and calculation are not consistent. Overlapping links were defined as "areas where the supplies of two different types of ES occur the in the same location on the landscape" (lines 371-372). Based on this definition, overlapping links should be not directional, i.e., the same for both ES link A to B and ES link B to A. But the presented calculations (Table 1) are directional, i.e., overlapping links differ between ES A to B and ES B to A. Also, they used different rules to calculate the overlapping weights for different ES pairs (lines 477-488), which made me confused. I do not understand why different rules should be applied to different ES pairs, and how would that affect the results. I wonder if those complicated steps were necessary, if we developed the calculation consistent with the definition (i.e., non-directional). (2) For topographic links, they only focused on the links having water flow (WF) as the source service. But ES other than WF can also be important ES source and form critical links. For example, different lands of plant agriculture (PA) can be connected by pollinators, which forms the connection PA-PA. This further points to problems with the calculation of topographic links. The weight of topographic link associated with WF was calculated as the summed WF values of the cells connecting two locations. That means they used related ES values to calculate corresponding ES link. This approach may miss important processes that truly connect ESs at different locations but poorly represented by the related ES values (e.g., pollinator movement are affected by land cover of forest and grassland) and would be difficult to derive useful mechanisms underlying the links.

---

## [Author Response]

Essential revisions:1) Most of the essential revisions address concerns regarding explanation and validation of the analysis approach raised by all three reviewers. Reviewer 1 points out that the number of ecosystem services studied is small and that it is important to delineate the limitations of the approach. Both other reviewers have questions about choices made for analysis which require clarification both in order to understand limitations, and also for the sake of reproducibility as well as for the potential uptake and application of such approaches by others.

We appreciate this thorough feedback and agree that addressing these issues will improve our manuscript. We detail below how we have done so for this resubmission.

1a. Some additional analyses must be conducted in order to assess the validity of the approach and its results for the chosen dataset.Reviewer 2: There is very little description on how (i) ecosystem attributes were mapped to ecosystem service supply, (ii) which data sources were used, why and what for and whether there are interlinkages between datasets that make the interpretation of the results less clear, and (iii) there is no description of the ES models – all in appendix but should be in the main text as this is the foundation of the functional connectivity analyses.

We used existing ES supply information from the cited companion report (Field et al. 2017: “Ecosystem Services Mapping to Support Environmental Conservation Planning Decisions in the Okanagan Region: Year 1 Progress Report”), which provides full details on original ES supply area mapping methods and data sources. However, we appreciate that readers of the current submission will benefit from a summary of this work being included in the main text. Therefore, we have moved and expanded some of the original full submission ‘Appendix 2’ information to the Materials and methods (see ‘Original ES supply data’ section) added text around dataset interlinkages (‘Original ES supply data’ section paragraph 4, lines 1242-1249); and provide a data sources tables in Appendix 6.

Reviewer 2: Functional connectivity is perhaps one of the most difficult aspects to validate, and landscape ecological literature has shown this for quite some time. Indeed, the authors mention the aspect of validation, but maybe it would be possible to compare the results with a null or a random walk model to see at least how comparable the results are?

Thank you for this suggestion – we agree that a null model-based validation is a valuable analysis to include in this study. Due to our large dataset, for this we have subset the study area to a single sub-basin which includes all 3 ES for this analysis and comparison. We have completed the validation analysis and added related information in Appendix 5.

1b. The authors should clarify the purpose of their analysis. Is this to discover mechanisms, or to evaluate the evidence for mechanisms which are hypothesized a priori?Reviewer 3: Although the authors aimed to provide answers to "what and where are the mechanisms responsible for maintaining these connections and, consequently, supporting the production of multiple ES" (lines 143-146). I did not get clear information on these "mechanisms" after the reading. These "mechanisms" seems were only first defined by the authors (the two types of links of ES) and then put into the calculation and mapping.

The purpose of our study has been clarified in the Introduction (lines 261-266), which is to “…identify relevant functional relationships between multiple ES and demonstrate how these can be spatially represented in the context of connectivity planning across a regional heterogeneous landscape.” This is reiterated in the Results (paragraph 1, lines 322-331), “…the focus of this study is not on how to produce the most accurate spatial representation of ES and their connections, but is on demonstrating a connectivity-based approach for visualizing and evaluating multi-ES relationships.” Our goal was neither to ‘discover’ nor to ‘evaluate evidence for… hypothesized’ mechanisms, and we have removed the original text in the Introduction cited by Reviewer 3 to avoid this confusion.

1c. The authors have also chosen an analysis method which may be especially tailored to water flow (WF) and less suitable for other variables, especially when considering their evaluation of topographic (versus overlapping) links.

We appreciate that our LCP-based method for topographic link creation will only be suitable for a certain subset of ES, and that not all pairs of ES will be functionally connected (i.e., neither topographic or overlapping link). We have addressed these issues as outlined below.

Reviewer 1: While an interesting proof-of-concept, it is clear that some ecosystem services are more likely than others to exhibit functional connectivity as described in this paper. For instance, the chosen waterflow regulation is necessarily a "flow" (thus exhibiting functional connectivity) while landscape aesthetics might not be.Reviewer 2: While yes LCP has been widely used in functional connectivity and hydrological studies, does this make it more likely to obtain certain types of results for waterflow regulation than for other ecosystem services? I would recommend the authors to discuss whether the choice of method for functional connectivity matters, and if there are certain methods that work best for one or another ecosystem services. The authors do acknowledge in the discussion that there are other methods, however, also justify the choice based on waterflow thus maybe making it more unlikely to find functional connectivity for other ecosystem services that may be more "corridor" or "resistance" based and thus would be better modeled with other methods?

We agree with the above assessments, have acknowledged the potential lack of suitability of our approach for certain ES in the Discussion section ‘Limitations and opportunities for future work’ (paragraph 4). A major task for future researchers applying our approach, as outlined by our Guidelines (Figure 6), will be to ‘Map functional connections’ between pairs of ES. This step will require future researchers to delve into the literature and find appropriate models specific to the functional mechanisms that connect specific pairs of ES. Identifying potential models specific to other pairs of ES is beyond the scope of our study, but we have expanded our discussion of other existing methods for spatially identifying and evaluating ES connectivity (‘Limitations and opportunities for future work’ paragraph 4).

Reviewer 3: For example, different lands of plant agriculture (PA) can be connected by pollinators, which forms the connection PA-PA. This further points to problems with the calculation of topographic links. The weight of topographic link associated with WF was calculated as the summed WF values of the cells connecting two locations. That means they used related ES values to calculate corresponding ES link. This approach may miss important processes that truly connect ESs at different locations but poorly represented by the related ES values (e.g., pollinator movement are affected by land cover of forest and grassland) and would be difficult to derive useful mechanisms underlying the links.

To apply our approach to the example provided by Reviewer 3, the specific ES supply areas to be connected (e.g., different PA supply and/or pollinator habitat areas) and the mechanism(s) to be represented would first have to be defined by the researcher. Then the most appropriate methods to spatially represent and weight these connections will have to be selected. We have expanded text around modelling approaches in Discussion section ‘Limitations and opportunities for future work’ paragraph 4, and have now included stepwise approach Guidelines for researchers (Figure 6). To discuss further the example provided by Reviewer 3, as animal pollination is defined as a regulating ES (e.g., https://ipbes.net/assessment-reports/pollinators), we would instead propose that pollinator habitat be represented as pollination ES supply area, which could encompass habitat areas both within and/or adjacent to PA areas. Then the functional mechanism would link PA and pollination services (rather than two separate PA areas), and could be represented by modelling pollinator movement between habitat areas (lines 929-934). We note that this does not address additional pollination services provided to wildlands for other, non-agricultural benefits. We clarify in the Discussion that, as long as the researcher clearly defines the functional mechanisms a priori and selects the appropriate model(s), our approach allows for any type of functional connection to be spatially represented (lines 934-939).

1d. It is not clear why the analysis of overlapping links is asymmetrical, or why this should be the case.Reviewer 3: Overlapping links were defined as "areas where the supplies of two different types of ES occur the in the same location on the landscape" (lines 371-372). Based on this definition, overlapping links should be not directional, i.e., the same for both ES link A to B and ES link B to A. But the presented calculations (Table 1) are directional, i.e., overlapping links differ between ES A to B and ES B to A.

We have clarified the definition of ‘overlapping links’ by expanding the text to include “… and a functional relationship exists between two ES” (Materials and methods section ‘Establishing functional connections between ES supplies’, paragraph 2, lines 1410-1412); and “Links can exist in one or both directions, with unique mechanisms operating from one ES to another” (Materials and methods section ‘Mapping and valuing ES supply patches’, paragraph 1, lines 1262-1263).

1e. It is not clear why the authors use different rules to calculate the overlapping weights for different ES pairs (lines 477-488), nor how such rules are derived (i.e., how should someone trying to reproduce this analysis derive the rules).Reviewer 3: I do not understand why different rules should be applied to different ES pairs, and how would that affect the results. I wonder if those complicated steps were necessary, if we developed the calculation consistent with the definition (i.e., non-directional).

The ‘rules’ applied to link weighting are based on the underlying functional relationships between two ES as defined by the researcher. These relationships, specific to the pairs of ES included in our study and to the direction of the relationship (e.g., WF exerting its functional influence on LA, which is unique from LA influence on WF), are outlined in Table 1. The definition and mapping of directional links is fundamental to the spatial representation and the valuation of links. We believe this issue is related to 1d above, and that clarifying the definition of ‘overlapping links’ (see 1d) addresses this.

2) What guidelines can the authors provide to generalize their proof-of-principle approach for future adopters and for others attempting similar analyses? To what extent is the chosen approach applicable to other ecosystem services than those three which the authors have chosen? Might other methods such as those brought up in 1), and mentioned by the authors, be more appropriate for other constellations of ecosystem services? Can the authors suggest how future analyses could incorporate ES dynamics?

We have addressed these helpful questions by expanding the text around alternative methods for other ES (‘*Limitations and opportunities for future work’* paragraph 4). We have also added guidelines for future adopters of our approach (Figure 6).

Reviewer 1: It is important to delve into the limitations of its approach, its potential replicability, the information needs, and how this limits its use. Would your methodological approach have problems with the incorporation of more ecosystem services? Static scenario (snapshot). In future research, could your methodological proposal overcome this difficulty? If not, how could it be overcome (please generate recommendations based on evidence and literature)?

We have included a discussion of limitations and potential solutions, including data requirements, increasing the number of connected ES, and dynamic network properties, in Discussion section ‘Limitations and opportunities for future work’. As mentioned above, we have also added guidelines for future research (Figure 6). We also discuss that our study represents a static example, and point to the potential for incorporating dynamic aspects of ES (‘Limitations and opportunities for future work’, paragraphs 3 and 4).

Reviewer 2: In Table 2 the authors propose the potential functional links (and the lack thereof) for their three ecosystem services. How robust is therefore the framework for ecosystem services for which we do not know whether or not there are functional connectivity links and how can we validate this approach?

We discuss that a large part of the task for future adopters will be to tailor the approach and the definition/modelling of functional connections to the based on scientific knowledge specific to the system and ES under study, and we provide some examples to help guide future work (‘Limitations and opportunities for future work’, paragraph 4). Related validation procedures will be specific to the mechanisms under study (e.g., Appendix 5; lines 925-928) and contemplating them in detail is therefore beyond the scope of our study.

3) Clarity and organization of text. Essential revisions under (1) and (2) already address this; however, I (the senior editor) note that the manuscript is already very long and although eLife generally does not impose strict length limitations, I do think this is an indication of some need for restructuring. Specifically, the introduction and discussion are disproportionately long, while essential details are missing from methods and results, as outlined above; and at the same time, some details which may be interesting only for specialists take up substantial space in the methods.

We agree and have reduced the Introduction, Materials and methods, and Discussion as much as possible based on Reviewer suggestions, without losing what we deem essential information. We have deleted some text and moved some to Appendices 2 and 7.

3a. Please streamline the introduction and the discussion to allow readers some mental space for the missing essential information. For the introduction, please consider what information readers need in order to understand the importance of your work and to evaluate your work specifically.3b. Appendices are a great tool to streamline the main text, but please consider that they should not be used not for information which is essential for most readers to understand your analysis and results. They should rather be used for important details that will be needed by specialists.Reviewer 2: I think in general the writing is very well done, and my main suggestion would be to reduce some unnecessary detail in the methods and in the discussion. More specifically:– Lines 251-259: do you need to detail the area covered by each land use type in such detail?

We have moved this text to the caption for Appendix 2 – Figure 1.

– Some descriptions of the methodologies are too detailed, and could be better elaborated in an appendix with the technical aspects. Keep in the main text the descriptions of the steps necessary for the LCP analyses and the identification of functional connectivity. I would suggest to move technical aspects on how the data was prepared for the LCP and of the deletion of segments is (in my opinion) too detailed.

We have moved some of this text to Appendix 7.

– Move limitations assessment to the main body of the text and explain how did you address/minimize/cope with these limitations within your analyses and results.

We have moved and expanded limitations assessment text to Discussion section ‘Limitations and opportunities for future work’, wherein we explain the relevance of these to our study, and discuss opportunities for how to address these.

3c. Regarding the discussion, I believe that addressing some of the concerns brought up in (1) and (2) will allow it to be more targeted. See also detailed comments from Reviewer 2.Reviewer 2: It would be important that the authors stay close to the what the results mean and their interpretation, that they discuss the implications of their choices, and would also recommend reducing some of the text of the discussion for a better focus. While there are four ways this study could be expanded to: (i) test whether LCP results differ from random or null expectations for the three ecosystem services, (ii) expand on the number of ecosystem services, (iii) test different functional connectivity metrics and models, and (iv) validate functional connectivity, I am aware that would be several additional studies. I would therefore suggest focusing a few analyses on the first point, to test whether the results presented herein are sensitive to the choices for ecosystem services and differ from random or null expectations [note from senior editor: see Essential Revision 1]. The other expansions could be, at the moment, discussed – for which I would recommend reducing and replacing some discussion content, which I will detail below.– Lines 711-727: maybe explain these results so that they don't repeat figure 8 – what is the most common LULC within all the pairs, does it differ between the topographic and the overlapping pairs? Does it differ between the pairs of ES? How much of these contributions are a function of LULC entering the models for ES and the weights attributed to LA, PA and WF?

We discuss our results from the perspective of LULC in Discussion section ‘*The case study: ES connectivity across a heterogeneous regional landscape*’, paragraph 3.

– Lines 879-914 – to me these are off topic and would greatly reduce to the main aspects that allow interpret your findings and implications. These can and should be replaced by a section on the limitations.

We have added discussion of limitations in Discussion section ‘Limitations and opportunities for future work’.

– Lines 921-948 and lines 950-967: again can be reduced – I understand you want to explain the implications of your findings but I would first focus on interpreting them, provide the limitations and what is that we learn from it, and then make some inferences of what implications might be – there seems to be several types of implications and maybe you can focus on (i) method and data implications – you already mention that this is a first analysis of ES functional connectivity which is quite new – but does it work for which data and for which method, and for which ES model?, (ii) ES research implications – this is very important – spatial relationships have been used to understand synergies and trade-offs and here you bring a new light to it – where could this lead to new knowledge about ES? and (iii) landscape and conservation planning implications – these are obvious if we want to maintain ES flows and conserve and plan development that does not break this flow. There are bits of each scattered throughout the discussion and would be helpful to consolidate these.

We hope the above has been addressed through expansion of limitations and opportunities for future work in the main text, general restructuring of the Discussion, and in the Conclusion.

– Line 1126: why focus on the visualization but not on the outcomes of what functional connectivity is? In general, I would focus the Discussion section more.

To clarify, the focus of our study is to provide an example of a new tool for visualizing functional connectivity between multiple ES, as we have now clarified in the Introduction (lines 261-266) and Results (paragraph 1, lines 322-331).